# Supervised mutational signatures for obesity and other tissue-specific etiological factors in cancer

Bahman Afsari[1†], Albert Kuo[2†], YiFan Zhang[2], Lu Li[2], Kamel Lahouel[1], Ludmila Danilova[1,3], Alexander Favorov[1,3], Thomas A Rosenquist[4], Arthur P Grollman[4], Ken W Kinzler[5], Leslie Cope[1], Bert Vogelstein[5], Cristian Tomasetti[1,2]*

[1]Division of Biostatistics and Bioinformatics, Department of Oncology, Sidney Kimmel Cancer Center, Johns Hopkins University School of Medicine, Baltimore, United States; [2]Department of Biostatistics, Johns Hopkins Bloomberg School of Public Health, Baltimore, United States; [3]Laboratory of Systems Biology and Computational Genetics, Vavilov Institute of General Genetics, RAS, Moscow, Russian Federation; [4]State University of New York at Stony Brook, Stony Brook, United States; [5]Ludwig Center & Howard Hughes Medical Institute, Johns Hopkins Kimmel Cancer Center, Baltimore, United States

*For correspondence:
ctomasetti@jhu.edu

[†]These authors contributed equally to this work

**Abstract** Determining the etiologic basis of the mutations that are responsible for cancer is one of the fundamental challenges in modern cancer research. Different mutational processes induce different types of DNA mutations, providing 'mutational signatures' that have led to key insights into cancer etiology. The most widely used signatures for assessing genomic data are based on unsupervised patterns that are then retrospectively correlated with certain features of cancer. We show here that supervised machine-learning techniques can identify signatures, called SuperSigs, that are more predictive than those currently available. Surprisingly, we found that aging yields different SuperSigs in different tissues, and the same is true for environmental exposures. We were able to discover SuperSigs associated with obesity, the most important lifestyle factor contributing to cancer in Western populations.

## Introduction

Cancer is the end result of a process of accumulation of genetic and epigenetic alterations. A small fraction of these alterations is inherited and the remainder is due to either random errors made during DNA replication or to environmental factors (*Mucci et al., 2016*; *Stadler et al., 2010*; *Stewart and Wild, 2014*; *Tomasetti, 2019*; *Tomasetti et al., 2017a*; *Tomasetti et al., 2017b*; *Tomasetti and Vogelstein, 2015*; *Tomasetti et al., 2013*). Delineation of the etiologic basis of these mutations not only can illuminate pathogenesis but has also immediate applications for prevention (*Song et al., 2018*).

Mutational signatures can provide unique insights into etiology because different mutational processes result in different types of mutations. For example, it has long been known that ultraviolet light often results in C to T transitions at dipyrimidine motifs (*Peng and Shaw, 1996*; *Saini et al., 2016*), while aging is associated with deamination at CpG dinucleotides (*Pfeifer, 2006*). Carcinogens such as tobacco smoking (*Govindan et al., 2012*; *Hainaut and Pfeifer, 2001*; *Imielinski et al., 2012*), aflatoxin (*Bailey et al., 1996*), and aristolochic acid (*Hoang et al., 2013*; *Poon et al., 2013*) are also known to induce characteristic mutations at specific motifs.

Based on these classical studies, systematic analyses of genome-wide sequencing data have been performed in an effort to discover new mutational signatures associated with various exposures. The classic and most commonly used approach (*Alexandrov et al., 2015*; *Alexandrov et al., 2016*; *Alexandrov et al., 2013a*; *Alexandrov et al., 2013b*) employs non-negative matrix factorization (NMF) (*Lee and Seung, 1999*). NMF is an unsupervised dimension-reduction machine learning technique, that provides a selected number of patterns without requiring any knowledge of the type of exposure (e.g. smoking, aging, sunlight) to which a cancer patient might have been exposed. In Alexandrov's implementation, the patterns are based on all possible 3-base-pair motifs, with the mutated base in the middle, and each pattern corresponds to a probability distribution of these 96 basic motifs. Once obtained, these patterns, termed 'signatures,' are retrospectively correlated with previously established mutation patterns or with known exposures to identify potential mutational processes underlying the signatures (*Alexandrov et al., 2015*; *Alexandrov et al., 2016*; *Alexandrov et al., 2013a*; *Alexandrov et al., 2013b*). These studies inspired a new way of assessing genomic data to derive insights about cancer etiology.

But can the approach of Alexandrov et al. be improved? It is true that the unsupervised approach does not require knowledge of an exposure to derive potential signatures. However, well-annotated clinical data are required to understand whether such signatures are associated with any environmental exposures. Interestingly, whenever such clinical data are available, *supervised learning* methods are expected to identify stronger associations and make more accurate predictions than unsupervised ones (*Hastie et al., 2009*). Moreover, future improvements in clinical annotation will improve the accuracy of the signatures obtained via supervised methods, but cannot improve unsupervised signatures because the latter are not formulated on the basis of annotated data. And, as we will show below, even when annotation for an exposure is not possible because the exposure itself is unknown, it is better to apply an unsupervised technique to detect the unknown patterns only after a supervised method has been used to remove from the data all the known, clinically annotated exposures.

We have developed a supervised algorithm to determine new mutational signatures, termed 'SuperSigs'. We then tested whether these supervised signatures would outperform previously described unsupervised ones in predicting the presence of various etiological factors in patients for whom both clinical and sequencing information was available. Finally, we determined whether new biological insights could be obtained from these new signatures, focusing on obesity.

## Results

### Supervised method for mutational signatures with low-variance features of variable length (SuperSigs)

To obtain our SuperSigs signatures, we analyzed sequencing data from 30 types of cancers recorded in The Cancer Genome Atlas (TCGA) database (see Materials and methods). Four key features distinguish our approach for identifying signatures.

1. Our primary methodological step is to use supervised machine learning, that is we learn the signatures from the data, by using the available annotation on clinical variables such as age, smoking status, and body mass index. By using this information explicitly, we expect to identify stronger associations and make better predictions.
2. We do not specify a pre-determined base length, such as 3-base pairs (*Alexandrov et al., 2013a*), as the fundamental unit of the mutational signatures. This provides greater flexibility because there is no reason to assume that all signatures are optimally described by the same base length units. In fact, a single signature may be defined on units of variable base lengths, featuring, for example, significantly elevated proportions of both C>A (i.e. a single-base substitution from C to A) and A[C>T]G (i.e. a single-base substitution from C to T with flanking bases A and G) mutations. We will restrict our present analysis to only 1, 2, and 3 base pairs lengths to simplify the presentation, as this is already sufficiently different from current methods using only trinucleotides, and leave the further extension to future work.
3. We employ a probabilistic approach to signature discovery. An important characteristic of any mutational process is its randomness. The mutational distribution caused by the same etiological factor varies greatly among exposed patients: a mutation type very frequent in some patients may not be common in others. From a biological point of view, it seems natural that

each patient – and in fact each cell - may have her/his individualized signature characterizing a specific etiological factor. Our signatures are therefore built only on a subset of selected features that are robust across the exposed population, that is features with relatively low variance, thereby increasing their predictive power.

4. We do not force the assumption that a given mutational process must have the same mutational signature across tissues, contrary to the approach developed by Alexandrov et al. where a given signature (e.g. signature 1) is the same across all tissues.

Our method for deriving mutational signatures is based on several steps. First, we construct a nested tree containing all potential features, with all mutations as the root, and all six single-base substitutions (C>A, C>G, C>T, T>A, T>C, and T>G) as the first level, followed by single-base substitutions with one flanking base as the second level, and by single-base substitutions with two flanking bases as the third level, and where the edges are placed between features which share mutations (*Figure 1*). In principle, our method can be applied to a tree with height greater than 3, by adding additional flanking bases, but here for simplicity and for comparing with current methods, we will only consider three levels.

After 'pruning' the tree in order to keep only the features that have counts significantly different from their expected values (one-sided binomial test with a 0.05 significance level, subject to Bonferroni correction), these remaining features are ranked based on their ability to classify a given exposure, that is to discriminate exposed patients from unexposed ones, as measured by the area under the receiver operating characteristic (ROC) curve (AUC). The set of n top features that provide the highest prediction performance in terms of AUC form our signature for a given exposure and are used for prediction (*Figure 1*). A detailed explanation of our method is provided in the Materials and methods section.

The value of a mutational signature can be assessed by its prediction accuracy (AUC) in classifying patients as exposed or not to the associated etiological factor, or by its correlation with exposure to that factor. In the next sections, we provide statistical evaluations for both, relying on the availability of clinical annotation for the etiological factor associated to that signature.

## Do mutational signatures add to prior knowledge about etiologic factors?

In addition to simple performance, it is also important to evaluate the degree to which a given mutational signature improves upon prior knowledge about the mutational effects of an exposure to an etiological factor (*Figure 2a*). For example, consider the case when clinical annotation is available and the main 'peak' of a mutational signature, that is its most common mutation, is already known before the mutational signature is obtained. The peak may be a nucleotide, a dinucleotide, or a trinucleotide, depending on the specific mutational process. For example, prior validated knowledge indicated that aging induces [C>T]G mutations, and smoking induces C>A mutations. The added value of a mutational signature then depends on the additional 'information' that that signature provides beyond this already-known peak. If a mutational signature yields additional mutations that, under the effects of a given exposure, the genome is enriched for—but was previously unknown to be—then that signature adds valuable information to prior knowledge. Mathematically, a mutational signature is represented by the set of 'weights' that that signature attributes to all mutations included in the analysis, with the larger weights associated to mutations the signature is more enriched for. If these weights enable a mutational signature to have a higher prediction accuracy, or correlation, than random weights do, then we say that that mutational signature provides 'information'.

To statistically evaluate the added value of the information provided by the signatures of Alexandrov and colleagues, hereafter termed 'unsupervised', as well as of our SuperSigs, we compared both of their performances against random alternatives carrying no additional knowledge beyond the known peak, for both aging and smoking (Materials and methods). We termed these prior knowledge signatures 'random' because they were purposely created to just reflect random noise around the already known peak (*Figure 2b*). Such random signatures are of course only meaningful when there is a peak that is already known and cannot be meaningfully constructed without prior knowledge.

We obtained sequencing data for thirty tumor types, from the TCGA Genomics Commons (https://portal.gdc.cancer.gov). After splitting each dataset randomly into training and test

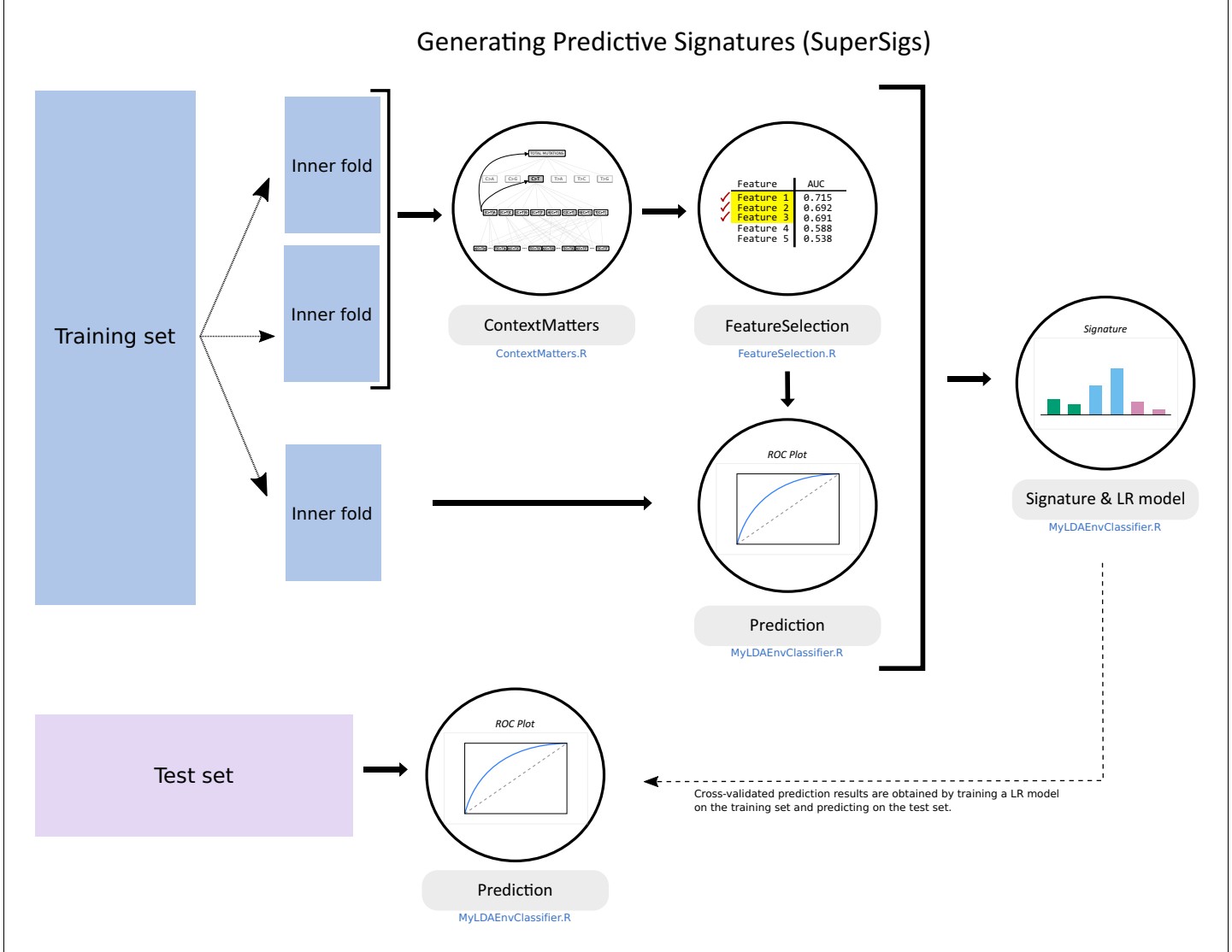

**Figure 1.** Flowchart of the supervised methodology for predictive mutational signatures. A schematic representation of the key steps contained in the supervised methodology. After splitting the TCGA dataset into training (80% of data) and test (20%) sets, 'ContextMatters' and 'FeatureSelection' are used to learn the candidate features. The final predictive features are then selected by learning the mutational differences between exposed and unexposed samples in the 'Prediction' step. These predictive features with their corresponding average rates derived during training form the supervised mutational signature (SuperSig), which is then used to predict exposure to an etiological factor in the test set (see Materials and methods for more details).

The online version of this article includes the following figure supplement(s) for figure 1:

**Figure supplement 1.** Supervised feature engineering.

partitions, we applied the method above to derive signatures of aging and smoking in the training data, evaluating performance in the test data. Our SuperSigs aging signatures, applied to classify patients in a binary fashion (i.e. young versus old) yielded a median AUC of 0.73, calculated over 30 tumor types, outperforming our random aging signature (single peak; median AUC = 0.63), which was built on the well-supported observation that over time, cytosines will consistently deaminate to thymine in the CpG context (*Figure 3a*, *Figure 3—figure supplement 1*, *Supplementary file 1*). When the signatures are used in a regression setting, to predict age as a continuous variable, the median correlation for SuperSig predictions was rho = 0.39 (*Supplementary file 1*). Our analysis on the same data yielded a median AUC = 0.57, and rho = 0.25, for the unsupervised aging Signature 1 (*Figure 3a*, *Figure 3—figure supplement 1*, *Supplementary file 1*). The combination of the 'clock-

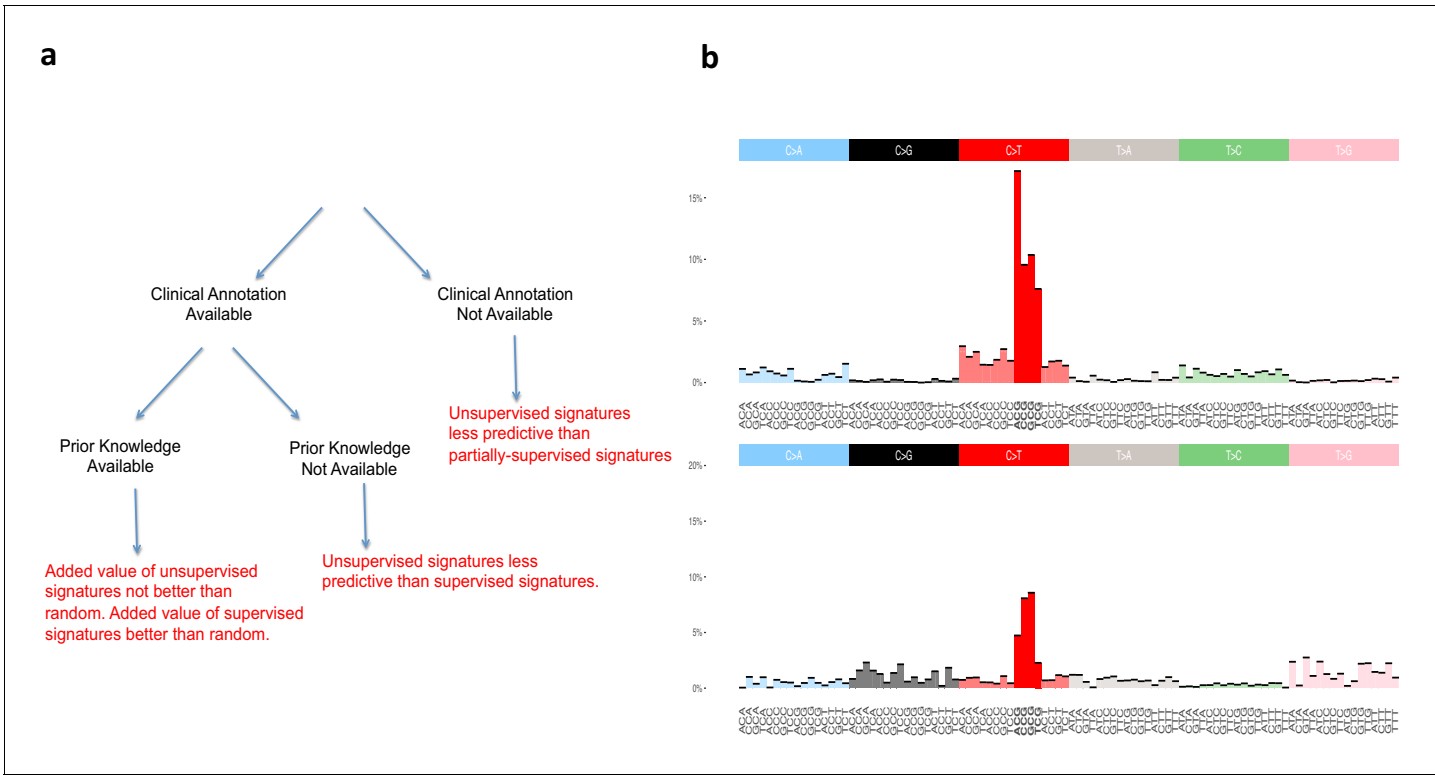

**Figure 2.** Supervised and unsupervised approaches to mutational signatures. (**a**) The three possible scenarios in which the supervised and unsupervised approaches can be compared (black) and a summary of each comparison (red). (**b**) Unsupervised versus random. The signature at the top of the figure is the unsupervised 'aging' Signature one from *Alexandrov et al., 2013b*. We want to assess the value of this signature beyond the 'peak' at [C>T]G (bold red color), that is we want to evaluate how valuable is the rest of the distribution (colors not in bold) as found by the unsupervised method. The signature at the bottom of the figure is an example of randomly generated single peak signatures based on sampling from a uniform distribution. Note that the normalized frequency of the mutation type corresponding to the peak of this randomly generated signature is not a fixed value; it happens to carry by chance the highest weight of the distribution over [C>T]G (bold red color) mutations among a set of 30 signatures generated randomly (see Materials and methods section for their construction).

wise' unsupervised Signatures 1 and 5 (*Alexandrov et al., 2015*) performed slightly better (median AUC = 0.61), although it did not improve on the random signature (*Supplementary file 1*). Unsupervised signatures for aging were not present in four of the tissues, while all tissues had aging SuperSigs.

We next evaluated the performance of these signatures with respect to smoking status across eight tissues known to be significantly affected by smoking and for which there was smoking status information in the TCGA database. Specifically, bladder (BLCA), cervical (CESC), esophageal (ESCAD and ESCSQ), head and neck (HNSCC), kidney (KIRP), lung (LUAD), and pancreatic (PAAD) cancers. The SuperSigs added value to prior knowledge while the unsupervised signatures did not, as the AUCs obtained by the SuperSigs were higher than the ones achieved by the random single peak but not so for the unsupervised ones (median AUCs for smoking: SuperSigs = 0.68, single peak = 0.55, unsupervised = 0.56) (*Figure 3b*, *Figure 3—figure supplement 1*, and *Supplementary file 1*). The correlation with smoking packs of the SuperSigs was 0.27 versus 0.23 when using the unsupervised smoking signatures. These results were confirmed with cross-validation, and even when using with the SuperSigs the same prediction method, non-negative least squares (NNLS), that was used by Alexandrov et al. (*Figure 3—figure supplement 1*, *Supplementary file 1*, and Materials and methods).

These data do not indicate that unsupervised signatures for aging and smoking are meaningless. However, the data indicate that the unsupervised signatures do not add any information to prior knowledge of a peak at [C>T]G for aging and at C>A for smoking. Optimally, an algorithm based on genome-wide cancer genomic sequencing data should add information that was not available from

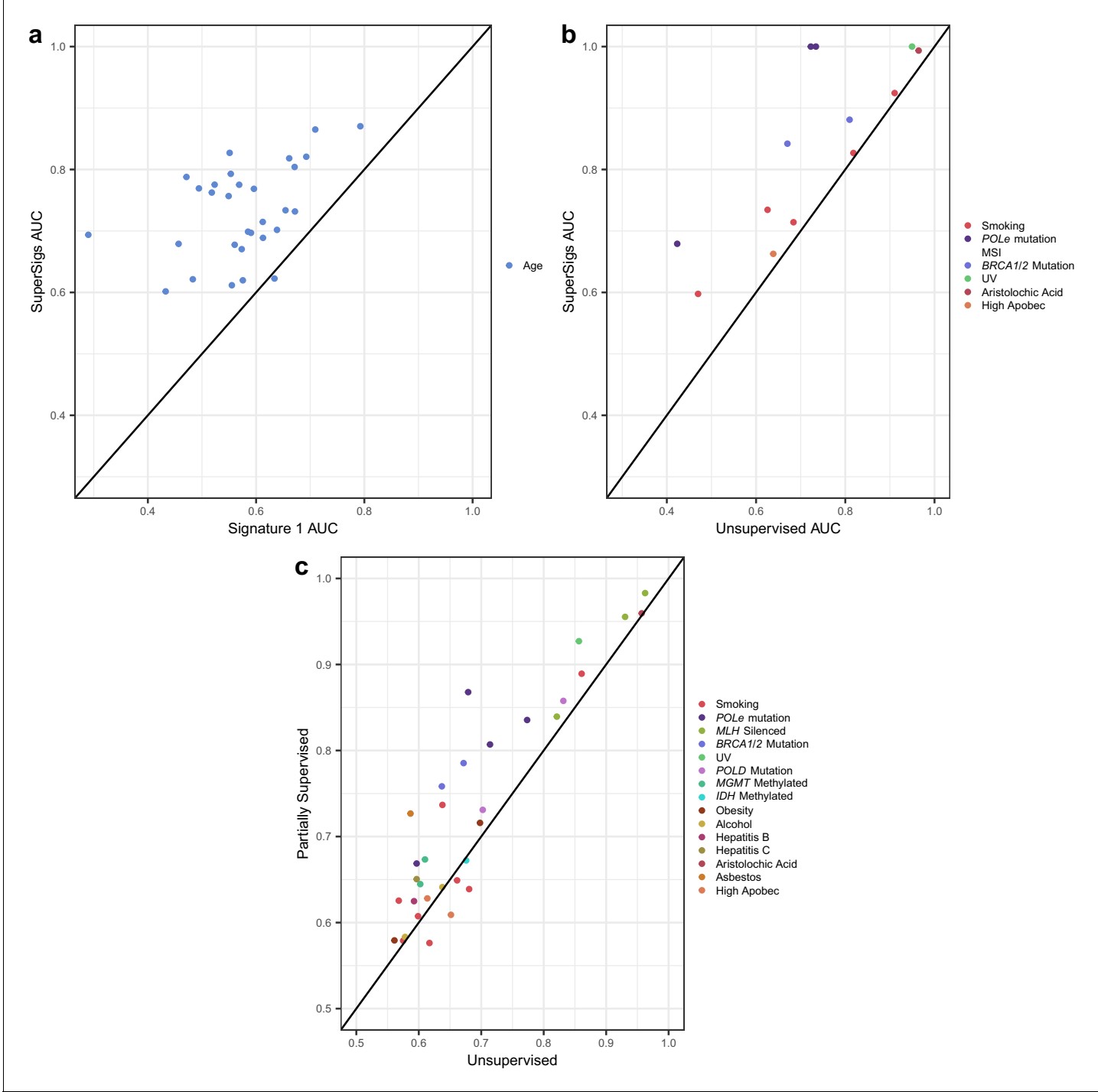

**Figure 3.** Comparisons of prediction accuracies (AUCs) of supervised, partially supervised, and unsupervised methodologies. (a) Supervised age SuperSigs vs unsupervised Signature 1 over 30 tumor types; (b) SuperSigs vs unsupervised signatures for all annotated etiological factors other than age found in *Alexandrov et al., 2013a*, in tumor types for which the unsupervised signature was present (for the full list see *Supplementary file 1*). (c) Partially supervised vs unsupervised NMF signatures for all annotated etiological factors other than age (see Materials and methods). Each combination of tumor type and risk factor (e.g. lung adenocarcinoma and smoking) yields a signature and is represented by one point, which depicts the prediction accuracies of the unsupervised approach (x-axis coordinate value) versus the supervised (a–b) or partially supervised (c) one (y-axis coordinate value). Apparent AUCs are reported. The great majority (c) or essentially all (a–b) points lie above or on the line, indicating the greater accuracy of the supervised and partially supervised approaches.

The online version of this article includes the following figure supplement(s) for figure 3:

**Figure supplement 1.** Unsupervised, random, and supervised methods' comparisons.

*Figure 3 continued on next page*

*Figure 3 continued*

prior studies, and SuperSigs indeed added such information that goes beyond the previously known mutational peaks (*Figure 2a*).

## Other comparisons between supervised and unsupervised signatures

Do supervised signatures perform better than unsupervised ones when no prior knowledge about an etiologic factor is available (second scenario in *Figure 2a*)? For factors (other than age) which could be evaluated by unsupervised methods, the median AUC of the unsupervised method was 0.77, while the median AUC for SuperSigs was 0.90 (*Figure 3b*, *Supplementary file 1*, Materials and methods).

Can we predict whether an individual patient was 'exposed' to a given etiologic factor simply from the SuperSigs in that patient's cancer genome sequencing data? In several cases, this was possible with high accuracy. For example, the cross-validated AUC was 0.90 when classifying patients with lung adenocarcinomas (LUAD) as smokers versus never-smokers. Similarly, the cross-validated AUC was 0.96 when classifying bladder urothelial carcinomas (BLCA) patients who were exposed or not to aristolochic acid (*Supplementary file 1*). At the same time, some exposures provided weaker performances: the cross-validated AUC was 0.62 when classifying patients with liver cancers (LICH) as drinking alcohol more than once per week vs. less than once per week.

When clinical annotation is not available for an etiologic factor (*Figure 2a*), the unsupervised method may appear to be the only viable approach. One limitation of any supervised approach is indeed that it cannot learn signatures of factors for which no annotation is currently available. And it may be desirable to have a method that is able to discover patterns of exposures, even when they are unknown. In this case, are we forced to use an exclusively unsupervised methodology? The answer is no. While it is true that our supervised methodology requires clinical annotation, there are many cases where we may have annotation available for at least some factors—for example patient's age is typically available for each sample—and not for others. In that case, that is any time some annotated factors are present in a sample, it is better to take care of them first, by identifying them using a supervised approach and removing their effects, to then apply an unsupervised methodology on the mutational 'leftover', rather than only using the unsupervised methodology on the whole. That is, we can first take advantage of our supervised knowledge of all exposures with available annotations. After learning those SuperSigs, we can 'subtract' their effects from the mutational load of the patients exposed to those annotated factors, and then perform an unsupervised analysis, such as non-negative matrix factorization (NMF), on the leftover, to investigate the presence of further mutational patterns. We term this approach 'partially supervised' and report its results in *Figure 3c*, showing that this method indeed achieves higher AUCs than the exclusively unsupervised approach (see also *Figure 3—figure supplement 36*). We provide the technical details of this partially supervised extension of our method in the Materials and methods section.

## SuperSigs for aging and other factors vary with tissue type

It has long been known that certain types of mutations, such as C>T transitions resulting from cytosine deamination, accumulate with age. We wondered whether other mutational signatures of aging were present in cancers and whether they varied among tissue types. To avoid confounding factors as much as possible, the analysis was confined to patients without known cancer-associated environmental exposures and without known germline predispositions to cancer.

We thereby obtained SuperSigs associated with aging for each cancer type analyzed, examples of which are shown in *Figure 4a* (*Figure 4—figure supplements 1–30*, and *Supplementary file 2* for the full set). Not surprisingly, we found C>T transitions to be present in large fractions in many cancer types. However, others, such as C>A transversions in stomach and prostate adenocarcinomas, and adrenocortical carcinomas, T>C transitions in liver hepatocellular carcinomas, C>G transversions in colorectal adenocarcinomas, head and neck squamous cell carcinomas, prostate adenocarcinomas, renal clear cell carcinomas, testicular germ cell tumors, and uterine corpus carcinoma, and any mutations of the T pyrimidine in prostate and kidney cancers, and testicular tumors, had not been previously described as major age-associated mutations (*Figure 4a*, and *Figure 4—figure supplements 1–30*).

We also sought to identify tissue-specific SuperSigs associated with specific environmental carcinogens. The analysis was performed after controlling for age and for other relevant covariates (Materials and methods). We obtained tissue-specific SuperSigs for smoking, alcohol, hepatitis B and C virus infection (HBV, HCV), aristolochic acid (AA), asbestos, and ultraviolet (UV) light (*Figure 4b*, *Figure 4—figure supplements 31–67*, *Supplementary file 2*, and Materials and methods). We also wanted to identify mutational signatures associated with defective DNA polymerization or repair, controlling for age, and other relevant covariates. We were thus able to obtain tissue-specific Super-Sigs for mismatch repair deficiency, mutations in DNA polymerase delta or epsilon genes, mutations in the breast cancer susceptibility genes BRCA1 or BRCA2, methylation of the MGMT and IDH1 genes, and APOBEC (*Figure 4b*, *Figure 4—figure supplements 31–67*, *Supplementary file 2*, and Materials and methods).

In several cases, the SuperSigs associated with the same mutational factors varied across tissues, just as they did with aging. For example, the SuperSigs associated with smoking were very different in bladder, head and neck, and lung cancers (*Figure 4c*). And the SuperSigs associated with BRCA gene mutations were considerably different between breast and ovarian cancers (*Figure 4—figure supplements 36–37*). There were, however, SuperSigs that did not vary much among tissue types, for example those based on mismatch repair deficiency, and some of those associated with inherited factors (*Figure 4—figure supplements 31–67*).

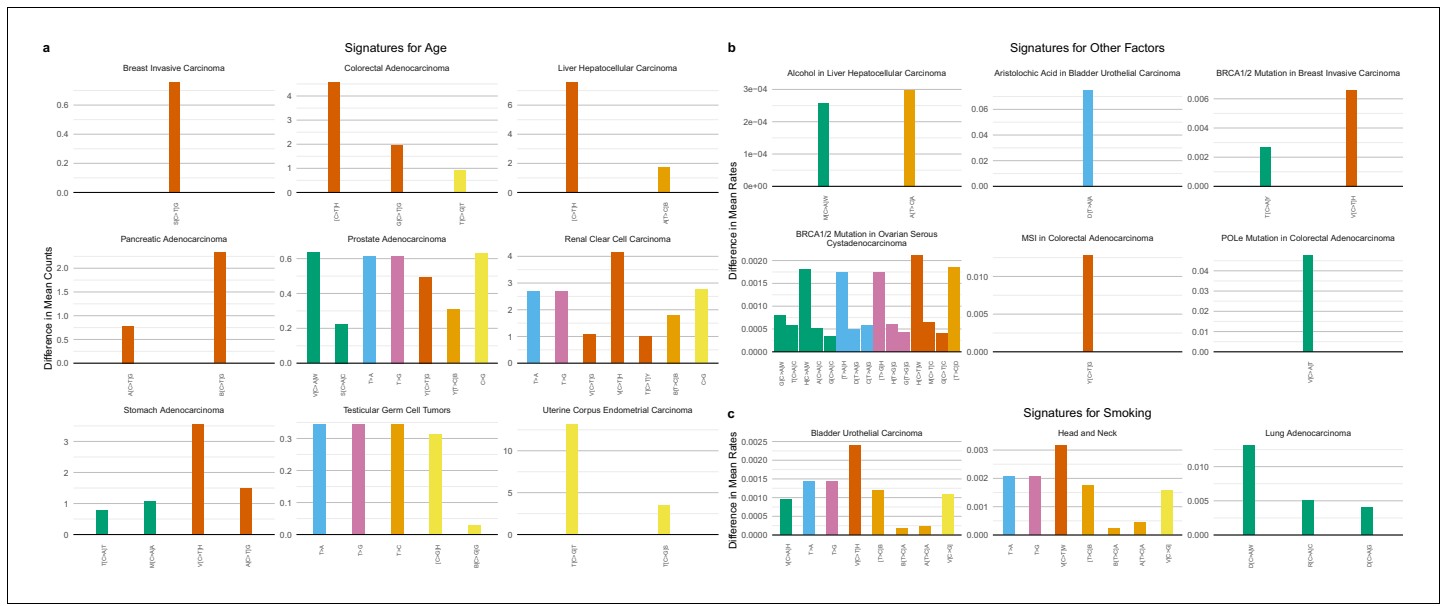

**Figure 4.** SuperSigs in various tissue types. All predictive features of a signature are depicted (IUPAC notations: B=not A, D = not C, H = not G, V = not T, W = A or T, S = C or G, M = A or C, K = G or T, R = A or G, Y = C or T). The color of each bar is representing the point mutation type as follows: C to T mutations = red, C to A = green, C to G = yellow, T to C = orange, T to G = purple, T to A = blue. The difference in the mean mutation count (for age) or in the mean rate (=mutation count/age, for all other exposures) between exposed and unexposed (old versus young for the age signature) is reported for each predictive feature. (a) Examples of age signatures. *Figure 4—figure supplements 1–30* and *Supplementary file 2* for the full list. (b) Examples of environmental, DNA polymerization or repair, and other factors' signatures. *Figure 4—figure supplements 31–67* and *Supplementary file 2* for the full list. (c) Examples of smoking signatures in different tissues. The three smoking SuperSigs presented here are the ones that achieved an AUC > 0.60 in cross-validation. See *Figure 4—figure supplements 59–66* and *Supplementary file 2* for the full list.

The online version of this article includes the following figure supplement(s) for figure 4:

**Figure supplement 1.** SuperSigs for age.
**Figure supplement 2.** SuperSigs for age.
**Figure supplement 3.** SuperSigs for age.
**Figure supplement 4.** SuperSigs for age.
**Figure supplement 5.** SuperSigs for age.
**Figure supplement 6.** SuperSigs for age.
**Figure supplement 7.** SuperSigs for age.
**Figure supplement 8.** SuperSigs for age.
**Figure supplement 9.** SuperSigs for age.
**Figure supplement 10.** SuperSigs for age.
**Figure supplement 11.** SuperSigs for age.
**Figure supplement 12.** SuperSigs for age.
**Figure supplement 13.** SuperSigs for age.
**Figure supplement 14.** SuperSigs for age.
**Figure supplement 15.** SuperSigs for age.
**Figure supplement 16.** SuperSigs for age.
**Figure supplement 17.** SuperSigs for age.
**Figure supplement 18.** SuperSigs for age.
**Figure supplement 19.** SuperSigs for age.
**Figure supplement 20.** SuperSigs for age.
**Figure supplement 21.** SuperSigs for age.
**Figure supplement 22.** SuperSigs for age.
**Figure supplement 23.** SuperSigs for age.
**Figure supplement 24.** SuperSigs for age.
**Figure supplement 25.** SuperSigs for age.
**Figure supplement 26.** SuperSigs for age.
**Figure supplement 27.** SuperSigs for age.
**Figure supplement 28.** SuperSigs for age.
**Figure supplement 29.** SuperSigs for age.

*Figure 4 continued on next page*

*Figure 4 continued*

Note that tissue specific differences with respect to etiologic factors are not possible to discover with the unsupervised approach described by Alexandrov et al. because the identity of a given signature across multiple tissues was a key theoretical assumption underpinning their approach.

We define the *mutational landscape* of an exposure in a tissue as the 96-long vector (96 trinucleotide mutations) where each entry is given by the average count of that mutation type, in the cohort of the samples with that exposure, divided by the average age in that cohort. The mutational landscape of aging is obtained in the same way using the cohort of samples without any known exposure ('unexposed'). Consider now the *distance* between any two mutational landscapes as given by the Pearson's correlation between the two mutational landscapes. The heatmap in *Figure 5* shows the 'closeness' - as measured by their correlation - between the mutational landscapes of any two cohorts of patients across all cancer types, clustering the more similar ones with each other (*Figure 3—figure supplements 2–18* and Materials and methods). The distances obtained by this alternative analysis indicate that the mutational landscapes produced by aging are spread all across the range, providing further evidence that the mutational processes associated with aging vary greatly with tissue type. This remained true even when subtracting the aging effect from the mutational landscape of the exposed cohort (*Figure 3—figure supplements 19–35* and Materials and methods).

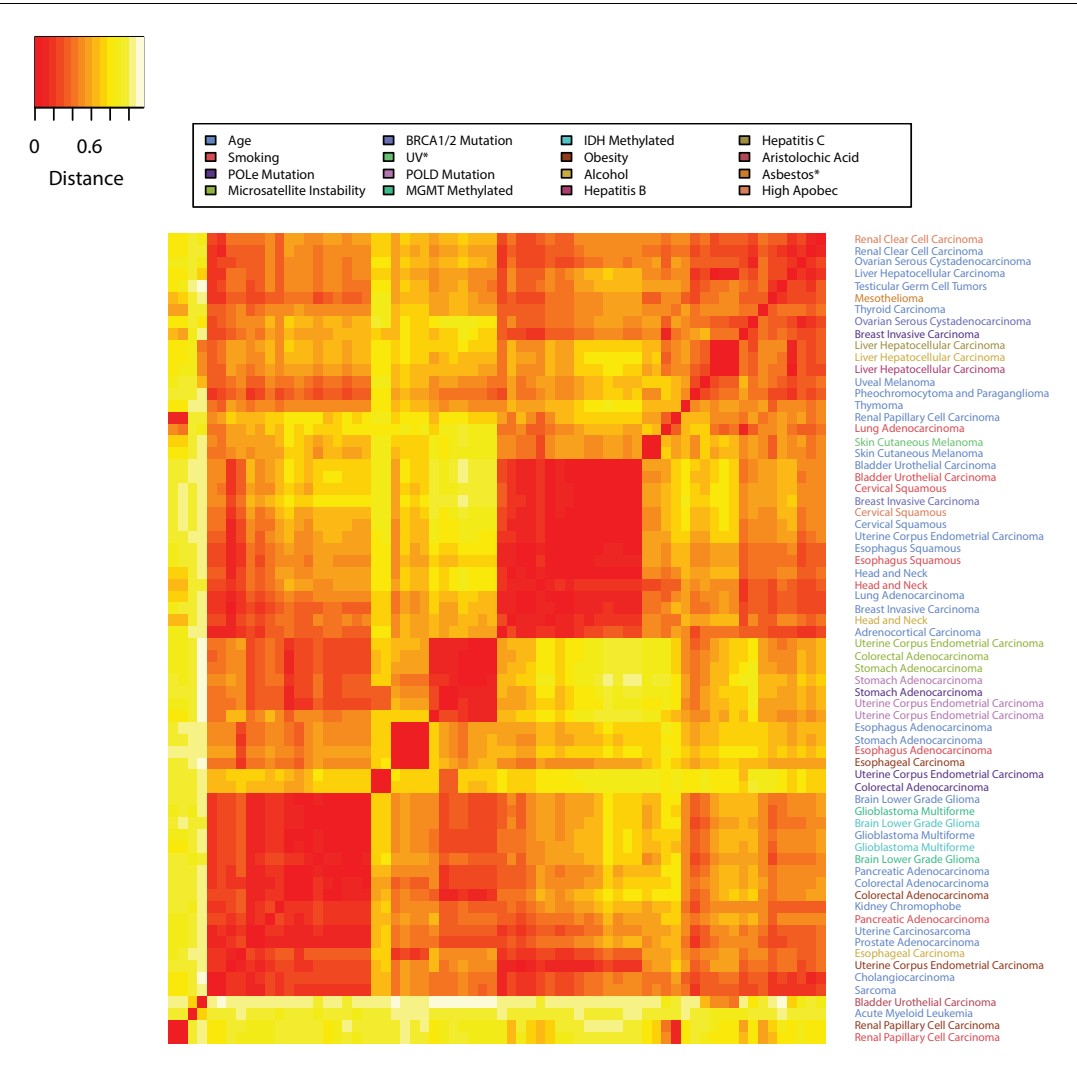

**Figure 5.** The tissue dependence of mutational signatures. Heat map of the distances among mutational landscapes of different etiological factors for different tissues. Pearson's correlation was used to calculate the distance (see Materials and methods). The lower the distance the more similar the corresponding mutational landscapes are.

Moreover, in several cases, the tissue-specific mutational landscape associated with an environmental factor was similar to the aging mutational landscape of the same tissue (*Figure 5* and *Figure 3—figure supplements 2–35*). For example, the mutational landscape in smokers was more similar to the aging one in the corresponding tissue than to the ones of smokers in other tissues (*Figure 3—figure supplements 2–35*). This again remained true for bladder, cervical, esophageal, and kidney cancers even when subtracting the aging effect from the mutational landscape of the exposed cohort (*Figure 3—figure supplements 19–35* and Materials and methods).

These analyses then suggest that a major effect of some environmental factors may simply be to increase the rate of cell division. This increase would induce a linearly proportional increase in mutation rate, but with a mutation pattern that remains similar to the one caused by normal aging, that is it would not be associated with new signatures such as those caused by direct interaction of carcinogens with DNA. Increases in the rate of cell division are known to occur when tissues are damaged or inflamed (*Cheah et al., 2015*; *Walser et al., 2008*). These observed similarities between the environmental and aging signatures then support the idea that, in certain tissues, the environmental factor's main effect is to induce inflammation in that tissue, thus increasing its cells' division rate.

## SuperSigs for obesity

Obesity (as measured by a body mass index, BMI, greater than 30) has emerged as the major lifestyle factor contributing to cancer in general (*Giovannucci et al., 1995*; *Hruby et al., 2016*; *Song and Giovannucci, 2016*). How obesity contributes to cancer risk, however, is unknown. For example, obesity could lead to cancer by inducing mutations or by stimulating the growth of neoplastic cells that have already acquired mutations (*Song et al., 2018*). If the former explanation were valid, there might be a mutational signature associated with obesity, but no such signature has been previously identified. Four cancer types associated with obesity in which adequate number of samples and body mass index data for a supervised machine learning approach were available: colon, esophageal, kidney, and uterine cancer. We were able to identify SuperSigs for obesity in all these cancer types (*Figure 4—figure supplements 49–52* and *Supplementary file 2*). Two of them, however, had an unreliable performance (AUC < 0.60) in cross-validation. For the other two (*Figure 6*), our ability to predict which patients were obese simply by the SuperSigs in their cancers – as measured by the apparent AUC – was 0.80 in kidney cancer (kidney renal papillary cell carcinoma - KIRP), and 0.66 in uterine cancer (UCEC) (*Supplementary file 1*). The obesity SuperSigs varied among the four cancer types, again suggesting the tissue specificity of mutational signatures associated with the same risk factor. The finding of a negative difference in the rate of T[C>G]T mutations in obese patients with uterine cancer (*Figure 6*) suggests an explanation for the observation that often the total number of somatic mutations found in cancers of obese patients is not significantly different from that of non-obese patients, when controlling for age. Only the mutational spectrum is different. Obesity could then induce interaction effects among mutational processes that go beyond the usual additive effects.

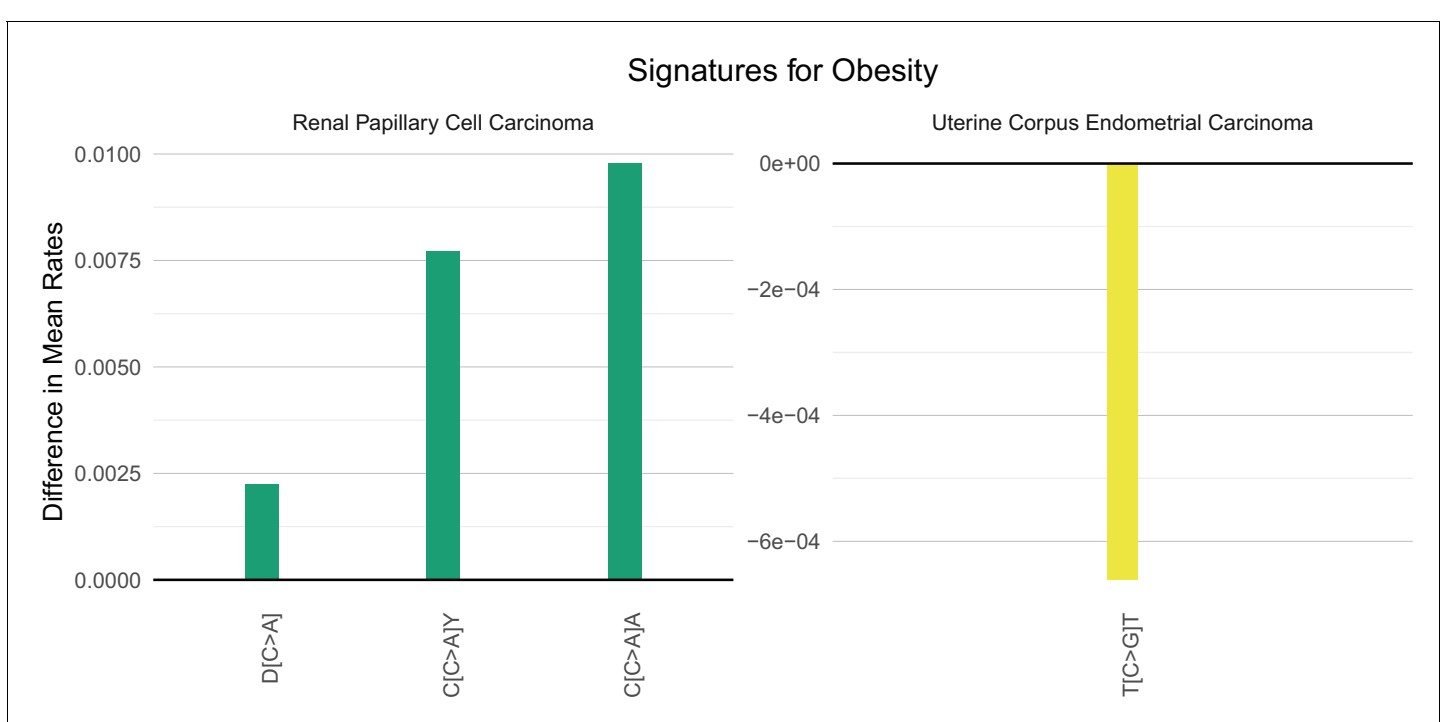

**Figure 6.** Mutational signatures of obesity in kidney (KIRP) and uterine (UCEC) cancer patients. All features of a signature are depicted (IUPAC notations: B=not A, D = not C, H = not G, V = not T, W = A or T, S = C or G, M = A or C, K = G or T, R = A or G, Y = C or T). The color of each bar is representing the point mutation type as follows: C to T mutations = red, C to A = green, C to G = yellow, T to C = orange, T to G = purple, T to A = blue. The difference in the mean mutation rate (mutation count/age) between exposed and unexposed is reported for each predictive feature present in the two mutational signatures for obesity. Bars falling below zero represent mutation types which are underrepresented when the given exposure is present.

## The proportion of mutations due to aging

We applied the supervised approach to estimate the proportion of the overall mutational load that can be attributable to normal aging rather than to other mutational processes. This proportion is directly provided by the contribution of the age SuperSigs in each patient (see Materials and methods). When considering all 30 tissues, we estimate that on average 69% of the mutations can be attributable to the normal endogenous mutational processes associated with aging, that is normal DNA replication (*Supplementary file 3*). This estimate is consistent with what previously reported in *Tomasetti et al., 2017b*. The proportion varied widely across tissues, for example it is 2% on average in endometrial cancers (UCEC) of patients with POLe mutations to 87% in pancreatic cancer (PAAD) patients who smoke. This estimated proportion may be an overestimate given the lack of full annotation for all environmental and inherited factors. At the same time, only non-annotated effects that are both common in the population of patients analyzed and that increase with age may erroneously end up in the age SuperSigs. Overall, the fact that the estimates for the contribution of the age signature are consistently large across the many tumor types analyzed points to a major role of the normal endogenous mutational processes.

## Validation of the SuperSigs

To validate our results, we tested the performance of the SuperSigs via four different analyses: cross-validation, random label shuffling, partial label switching, and validation on external data. First, we performed five iterations of 5-fold cross-validation of our SuperSigs (Materials and methods and *Supplementary file 1*). The cross-validation confirmed the positive performance of the SuperSigs as we have already reported in the previous sections. Overall, there was a drop in the median AUC of only 3 and 4 percentage points for the SuperSigs of age and of all other exposures, respectively, indicating a lack of major overfitting. As expected, there were a few signatures whose apparent AUCs were >0.60 but cross-validated AUC were <0.60 (6 out of 30 for age and 10 out of 37 for other exposures, see *Supplementary file 1*), and those should not be considered reliable. We then performed another test against overfitting, by randomly shuffling all the labels and then re-running the entire framework from feature selection to test prediction. Rather than obtaining similar AUCs between the shuffled and unshuffled datasets, the AUCs dropped to 0.50 in cross-validation on the shuffled data, providing evidence that our SuperSigs do not suffer from major overfitting (*Supplementary file 4*). Next, we tested the robustness of our methodology by considering scenarios where 5, 10, 20, and 25% of the clinical annotations for all etiological factors used in the training set were mislabeled. Even so, our performance in terms of AUC remained higher than the unsupervised one even with the highest percentage of mislabeling (25%; see the 'Robustness analysis with respect to mislabeling' section in the Materials and methods and *Supplementary file 5*).

Finally, we externally validated our results on an independent whole-genome sequencing (WGS) dataset from the International Cancer Genome Consortium (ICGC) database (downloaded from https://dcc.icgc.org/releases/PCAWG/). Only ICGC datasets that were not present in the TCGA database were used. All signatures that had previously achieved a cross-validated AUC of at least 0.6 on TCGA data were tested, as the others should not be considered reliable. Using, for each exposure, the associated logistic regression model trained on TCGA data only, and containing its SuperSig, we predicted the unexposed versus exposed status for all available factors and tissues in the ICGC data. The AUC for every of those signatures is reported in *Table 1*. We found that the predictive power of the SuperSigs observed in the TCGA dataset was retained when predicting on ICGC data, with relatively modest drops in AUC for a few signatures. The only exception was the age signature in melanoma (SKCM). This signature, however, had a borderline performance already in the original analysis (AUC = 0.61), probably due to the inability of the methodology to properly remove sun exposure as a confounding. At the same time, there were several SuperSigs that achieved better accuracies in the external validation ICGC dataset than in the original cross-validated TCGA, with gains up to 16–18 percentage points for the age signatures in ovarian (OV) and prostate (PRAD) cancers.

**Table 1.** External validation of the SuperSigs using the ICGC database.

Cross-validated performances (AUCs) of the indicated SuperSigs on TCGA data, compared to their performance when then used as predictors on ICGC data. The number $n$ of samples tested for each combination of tumor type and factor is indicated in parenthesis.

| Tissue | Factor | TCGA | ICGC |
|--------|--------|------|------|
| CHOL | AGE | 0.73 (n = 26) | 0.66 (n = 35) |
| HNSCC | AGE | 0.73 (n = 120) | 0.80 (n = 9) |
| KIRC | AGE | 0.81 (n = 123) | 0.75 (n = 82) |
| LIHC | AGE | 0.70 (n = 57) | 0.66 (n = 208) |
| OV | AGE | 0.71 (n = 87) | 0.87 (n = 92) |
| PAAD | AGE | 0.65 (n = 35) | 0.66 (n = 203) |
| PRAD | AGE | 0.65 (n = 305) | 0.83 (n = 120) |
| SKCM | AGE | 0.61 (n = 82) | 0.45 (n = 47) |
| STAD | AGE | 0.66 (n = 176) | 0.64 (n = 21) |
| LIHC | ALCOHOL | 0.62 (n = 154) | 0.66 (n = 25) |
| HNSCC | SMOKING | 0.81 (n = 354) | 0.78 (n = 13) |

## Discussion

The results recorded above lead to several important conclusions. First, supervised machine learning led to new signatures for a variety of etiological factors. These new SuperSigs are better at predicting an exposure than the signatures derived from unsupervised learning. And even when annotation is missing, the partially supervised extension of our method better predicts the underlying exposure than the unsupervised one. Overall, the above results indicate the clear advantages of the supervised approach. In addition, there is a well-known difficulty in choosing the correct number of patterns in any unsupervised methodology (see the 'The effect of model misspecification on the unsupervised signatures' section in the Materials and methods and *Figure 3—figure supplement 37*).

A second observation is that the SuperSigs usually varied with tissue type. In the majority of previous studies of signatures, it has been assumed that a specific mutational process produces the same signature in all tissue types (*Alexandrov et al., 2015*; *Alexandrov et al., 2016*; *Alexandrov et al., 2013b*; *Alexandrov et al., 2013a*; see *Blokzijl et al., 2016*; *Hoang et al., 2013* for exceptions). In contrast, SuperSigs were usually tissue-specific. The fact that the same risk factor, such as alcohol, might give rise to different signatures in different tissues might be viewed as surprising given historical views of exogenous carcinogens such as UV light. However, recent studies have suggested that tissue-specific differences in chromatin organization might underlie the tissue specificity of mutations, at least during aging (*Polak et al., 2015*). Moreover, the tissue-specific nature of SuperSigs is consistent with the tissue specificity of cancer predisposition syndromes. For example, inherited mutations in the fundamental genes involved in DNA repair or recombination, such as BRCA2, might be expected to result in predispositions to cancers of all types, but they only increase cancer risk in a limited subset of tissues. Our results suggest that the SuperSigs associated with BRCA2 indeed vary with tissue type. Clinical observations like these, together with the SuperSigs described here, support the idea that the nature of mutagenesis is highly dependent on tissue type, and often related to inflammation, which is – for example – known to be linked to obesity, suggesting important avenues for future research.

We were able to define a total of 67 SuperSigs but at most 2–3 of these SuperSigs appear to play a role in any single cancer. This stands in contrast to the widely used signatures discovered through unsupervised learning techniques. Even after eliminating unsupervised signatures that are present in a cancer but determined to be not 'significant' (*Alexandrov et al., 2013b*) and excluded from the analysis of a given cancer type, there are multiple instances where each of these remaining unsupervised signatures is found in essentially every cancer patient. For example, Signature 3, a signature for BRCA1 or two mutations, was found in virtually every breast cancer patient sequenced in TCGA

(see Figure S32 in *Alexandrov et al., 2013a*), whether the cancer had any relationship to the BRCA pathway or not. Similarly, Signature 4, a signature for tobacco smoking, and Signature 6, a signature associated with defective mismatch repair mechanisms (MMR), was found in virtually every liver cancer patient (see Figure S43 in *Alexandrov et al., 2013b*), though it is unlikely that all kidney cancer patients included in the TCGA database were smokers, and MMR-deficiency is rare in liver cancers.

A limitation of our method, and of any other method, is the quality of the clinical data currently available as well as the limited knowledge of the etiological factors to which patients are exposed. With respect to the quality of data we have tested the robustness of our methodology by considering scenarios where 5, 10, 20, and 25% of the clinical annotations for all etiological factors used in the training set were mislabeled. Even so, our performance in terms of AUC remained higher than the unsupervised one. Moreover, the results using the partially supervised method extension provide evidence that the clinical annotations for age and smoking status are already of sufficient quality to allow the partially supervised method to outperform the unsupervised one. And more sophisticated extensions to SuperSigs obtained by borrowing advanced sparse techniques used in, for example, the sigLassso method (*Li et al., 2020*), may provide further improvements.

There is currently much interest in performing genome-wide sequencing studies on very large numbers of cancer patients in whom clinical data are well-annotated. As such studies proceed, and as the knowledge of etiological factors advances, the power of the supervised learning approach described here will progressively increase. Notably, because unsupervised signatures are not based on such data, their power will not improve. We anticipate that the use of SuperSigs will therefore lead to accurate estimates of the fraction of mutations attributable to each specific environmental, hereditary, and replicative factor. Conversely, in certain cohorts, this approach could lead to the detection of a sizable fraction of mutations that cannot be attributed to any known source, potentially leading to new insights into pathogenesis, and in particular, avoidable pathogenic agents.

It may be argued that it is simpler to directly ask patients whether they are smokers or not, or what is their BMI, rather than using supervised signatures to predict it. We disagree with this point of view for two reasons. The main goal of using mutational signatures is not to predict an exposure, but to increase our understanding of cancer etiology and tumor evolution, that is to learn what the biological effects of different exposures are at the molecular level. This cannot be done just by asking the patient. Also, for many exposures – in fact all of them – the patient can only provide partial information like their BMI right now, or whether they drink alcohol or not. But exposures are much more complicated than that: are patients able to recall their lifetime changes in BMI, alcohol consumption, and so on? What can they tell us about their lifetime sun exposure beyond a summary statistic? Supervised signatures are designed to overcome that problem as they look for the molecular signal left by an exposure as it accumulated over time.

A final conclusion relates to obesity. Obesity is now considered the primary environmental risk factor for cancers in general, and with its increasing incidence, the number of cancers impacted by it is huge (*Giovannucci et al., 1995*; *Hruby et al., 2016*; *Song and Giovannucci, 2016*). Yet, the mechanisms underlying the effects of obesity on cancer risk are unknown. Numerous speculations about mechanism have been proposed, such as the effects of putative adipokines and a variety of other hormones or circulating metabolites on cell growth. Sequencing data analyses seemed to point away from a possible mutational mechanism, because often the total number of somatic mutations found in cancers of obese patients is not significantly different from that of non-obese patients. The discovery of SuperSigs for obesity in the four tissues analyzed – with two of them relatively robust in cross-validation – suggests that, at least in those tissues, part of the risk from obesity may be attributable to mutagenesis. This observation thus leads to specific testable hypotheses that can advance the field. For example, what circulating molecules in obese patients increase the accumulation rate of certain mutation types, giving rise to the SuperSigs described here?

## Materials and methods

### Data preparation and integration

We downloaded somatic exomic mutational data from the TCGA Bioportal (https://portal.gdc.cancer.gov) and filtered out the mutations which have less than 5% Variant Allele Frequency (VAF). Out of the total thirty-three datasets available, large B-cell lymphoma (DLBC) was not included in the

analysis because of the small number of samples available, while lung squamous cell carcinoma (LUSC) and mesothelioma (MESO) were excluded because of the extremely small number of patients unexposed to smoking and asbestos, respectively. For ovarian cancer (OV) and acute myeloid leukemia (LAML) whole genome sequencing data were used. The human genome reference build *hg38* was used to determine the context (flanking bases) for each mutation. The clinical information was downloaded from the website Cbioportal (http://www.cbioportal.org). For calculating the background frequency of each trinucleotide on both the exome and the genome, we used the R package, *deconstructSigs*. For the Unsupervised Signature method (*Alexandrov et al., 2013a*), we downloaded the signatures from the Cosmic Signature website (http://cancer.sanger.ac.uk/cosmic/signatures) and used the table http://cancer.sanger.ac.uk/signatures/matrix.png in order to determine which signatures were present in which tissue.

All analyses were performed using *R* version 3.5.2. Logistic regression was performed using *glm* from the *STATS* package. LDA was performed using the function *lda* from the package *MASS*. Non-negative matrix factorization (NMF) was performed using the function *nmf* with method 'Lee' from the package *NMF*.

## Filtering of the samples

To reduce the effect of confounding factors, we applied several filtering criteria. In each tissue type, we divided samples into two categories: (1) 'unexposed', meaning that no exposure to a known environmental factor was recorded, according to the available clinical annotation, and (2) 'exposed'. To mitigate the effects of other unknown factors in the unexposed group, we removed any sample with a mutational load more than 3 times higher than the median number of mutations found among the unexposed samples. We also excluded samples if the total number of mutations was equal to zero on the exome, a probable indication of low neoplastic cell content. We removed samples with microsatellite instability (MSI) or with a mutation in POLE/POLE2/POLE3/POLE4 or POLD1/POLD2/POLD3/POLD4 genes - except for when the signature for the specific effects of those mutations was the objective of the analysis - because of the known large increase in the number of mutations they induce. A tissue type was divided into subtypes whenever possible. Acute Myeloid Leukemia (AML) patients younger than 40 years old were not considered because we were interested in adult rather than childhood AML. Among the 'exposed' samples we excluded samples with known multi-factor exposures to minimize confounding factors and only evaluated samples with a single known exposure. Samples with unknown exposure were treated as unexposed.

## Measuring mutations

Mutation counts are used to characterize mutational burden when considering predictors of aging. For all other exposures, mutation rates (i.e. counts/age) are used. In a patient exposed only to time, that is unexposed to any known environmental or inherited factor, the rate of a mutation type is expected to remain constant irrespective of age – as dictated by the aging signature – while the absolute count is expected to increase with age. In contrast, in a patient exposed to an environmental or inherited factor, the rate of a mutation type as well as the count may change with respect to the age signature.

## Supervised methodology for generating signatures (SuperSigs)

We provide here the details for the method we developed to obtain the supervised mutational signatures (see flowchart in *Figure 1* of the main text).

At its simplest, a mutational signature of exposure is nothing more than a set of substitutions that characteristically occur at different rates in exposed tissue than in unexposed tissue. In practice, though, a few considerations suggested by prior biological knowledge quickly turn a simple calculation into a complex engineering problem. Specifically, a key principle of the SuperSig approach is that signatures may not be optimally described by the same base length units. Accordingly, we consider all single-base substitutions, with or without the flanking context bases, as potential, signature features. In addition to six single base substitutions: C>A, C>G, C>T, T>A, T>C, and T>G, named according to the pyrimidine of the mutated Watson–Crick base pair, there are 48 dinucleotides, in which the substitution is paired with a specific base as a prefix or as a suffix but not both (e.g. A

[C>T] or [C>T]G), as well as 96 trinucleotides (e.g. A[C>T]G), which include both flanking bases as context. Hence, we have a list of 151 *potential features* (6 + 48 + 96 + 1).

The resulting flexibility carries a price, however, as features are no longer independent. The simple substitution C>T spawns dinucleotide *children*, such as A[C>T], and trinucleotide *grandchildren* like A[C>T]G. Frequent, exposure-driven A[C>T] substitutions would increase the observed rates of both the C>T parent and the trinucleotide children, making it difficult to assign ownership to the correct generation. The next section, **ContextMatters,** describes our approach to solving this problem, while the sequel, **FeaturesSelection,** describes how candidate signature features are combined to create a final signature.

## Supervised feature engineering (ContextMatters)
### The mutational family tree
The set of features described above thus form a family tree, in which the observed mutational rate (or count, when learning the mutational signatures of aging) for each substitution is propagated down the tree to children and grandchildren (*Figure 1—figure supplement 1*). For completeness, the tree is augmented with a single root, *Total Mutations*, parent to all six simple substitutions, describing the overall mutation rate (or count, for aging). Such a tree can represent the mutations found in a single sample, or summarize results observed across a set of samples. In practice, we build two trees for each combination of exposure and tissue, to capture mutation rates separately in exposed and unexposed individuals, and combine them later.

All criteria for splitting samples in young vs old, smoker vs never-smoker, etc, were selected prior to any application of our algorithm or cross-validation. For the age analysis, we divided the unexposed samples into three groups (younger, middle-aged, older) based on the tertiles of the age distribution and discarded the middle group before training the algorithm (*Figure 3—figure supplement 39*). For all other exposures, unexposed and exposed – as indicated by the available annotation - formed the two groups except for: (1) obesity, where a BMI = 30 was used as the splitting value between obese and non-obese and (2) ultraviolet light (UV) and asbestos, for which samples with respectively the lowest 10% and 33% of the Total Mutations count were used as a proxy for the unexposed group, and all the other samples for the exposed one. We used the counts of the predictive features for age and the rates (=count/age) of the predictive features for all other exposures, over the two labeled groups. For age, the middle-aged group was excluded from the test set.

### Feature selection
Features of interest are selected in each tree by a two-phase process, first working down the tree from the root and then back up again. The very simple principle behind the first phase is that the mutation rate for each feature is to be compared to that expected by chance alone, to distinguish features that may be associated with exposure. As an unfortunate consequence of the family structure, however, the simplest implementation of this principle is biased toward the selection of late-generation features, where the propagation of individually insignificant deviations across 2 or 3 generations may add up to a significant cumulative difference. Thus, in practice each feature must pass a series of tests against a hierarchy of conditional null distributions defined by accounting for the observed mutation rates of each ancestor in turn. In consequence, unless proven otherwise, the mutational wealth of a given feature is explained by inheritance from its ancestors. This leads to the second phase of the process, where we work back up the tree, reevaluating all parent-child pairs selected in the first phase to make sure that we have not over-corrected, and erroneously attributed later generation wealth to earlier generations. Mathematical details are provided below.

### Phase (1) going down the tree
The hierarchy of conditional nulls is perhaps best described by example. If chance alone is at work, the expected number of C>T mutations would be *Total Mutation_Count * Normal_Frequency_of_C * 1/3*, the last factor accounting for three, equally likely substitutions for C. The C>T substitution would be selected as a candidate feature if the observed number of C>T mutations were significantly greater than the expected value, according to a one-sided binomial test. Moving down a generation, [C>T]A, as the child of the C>T substitution, and the grandchild of the total number of mutations (*Total Mutations*), would be tested twice to see if it significantly exceeded its expected

number based on the total number of mutations as well as the number of C>T. The expected value of [C>T]A mutations would be given by *Total Mutation_Count * Normal_Frequency_of_C * 1/3 *X*, where X is the expected frequency of CA (i.e. C followed by an A) out of all C nucleotides in the exome, as estimated by *deconstructSigs* (**Figure 1—figure supplement 1**).

The binomial test was based on an estimate of the sum of the number of mutations observed for that potential feature across all training samples, and the probability of success was set equal to the frequency of that potential feature, as expected by its representation on the exome. Specifically, the estimate of the sum of the number of mutations observed for that potential feature across all training samples was calculated by a bootstrap (100 times) for the sum of the pseudo count of that feature, of which we then took the median. The start for the pseudo count of the *Total Mutations* is set at 1000. For any other feature, the pseudo count starts from the proportion of that feature with respect to the exome, multiplied by 1000. We applied rounding to the outcome.

All results were considered significant at a p-value of 0.05, subject to Bonferroni correction for 150 tests, as *Total Mutations* is not tested against. If the null hypothesis was rejected, we selected that potential feature as a '*first-phase' candidate feature* for the next supervised selection step. First-phase candidate features are colored in grey in **Figure 1—figure supplement 1**.

### Phase (2) going back up

Once a list of first-phase candidate features had been thus selected, we pruned this list resulting in a smaller set of *second-phase candidate features* (colored in yellow in **Figure 1—figure supplement 1**). We did this by 'going up the tree', that is, by re-evaluating the significance of first-phase candidate features that are parents of first-phase candidate features. Indeed, some parent features may have been selected only because their children had higher than expected frequencies. We tested the parent by removing the contributions in terms of number of mutations present among the selected children to see if the count of the leftover in that parent would still be significantly higher than expected by chance. If it were, then that parent remained in the list as a second-phase candidate feature. And, for each sample, its mutation count will be updated by removing the mutations of the second-phase candidate feature children. Instead, if not significant, the parent was eliminated as a feature in that particular analysis. We named '*remaining mutations*' the feature containing the leftover of the *Total Mutations* and keep it as a second-phase candidate feature, to protect us from discarding important correlations that may not be tested by our algorithm.

### Combining partitions

For every factor other than age, we applied the above feature-engineering (ContextMatters) step separately to samples from patients that were respectively unexposed or exposed to the factor under consideration. We then combined these two lists of second-phase candidate features, which are both partitions, by considering all intersections and relative complements of the elements in the two original partitions, to form the minimal refinement of the two, and define this final list as the list of *candidate features*.

For aging signatures, we applied the feature engineering steps described above only to samples from patients who were unexposed to any known environmental or inherited factor. Therefore, we skip this step of combining partitions, because we only have one partition, that is its second-phase candidate features, which automatically provided its 'candidate features' list.

### Supervised feature selection (FeaturesSelection)

We performed the supervised feature engineering step described in the previous section in an inner loop of 5 iterations of 3-fold cross-validation. Each feature in each fold was ranked according to its ability to discriminate exposed samples from unexposed, based on the rates for that feature (or counts, as appropriate for the exposure). Discriminatory performance was measured by the area under the receiver operating characteristic (ROC) curve (AUC).

The fifteen lists of 'candidate features' obtained across the 3 folds of each of the five iterations was then combined by considering all intersections and relative complements of the features in the original lists. The minimal refinement thus obtained constitutes the final list of candidate features. These candidate features were then ranked by their median AUC across the 15 folds.

Also, for each fold, the number *n* corresponding to the top n ranked features that provided the highest AUC - using a multivariate, logistic regression classifier (LR) - was selected. The median of those 15 *n* values, defined as *n\**, was then used to select the first n\* candidate features according to their median AUC ranking. Out of these n\* features, the ones for which the median AUC $\leq 0.6$ were discarded, and the remaining features are what we defined as the *predictive features* for a given exposure.

### Signature representation (Signatures)

The set of predictive features selected above form the supervised signature (SuperSig). Two values are associated to each one of these predictive features: (1) the difference in mean counts (age) or rates (all other exposures) between the exposed and unexposed cohorts, and (2) the beta (β) coefficient for that feature as estimated by logistic regression. Both vectors yield critical information.

The difference in means for each feature, which is the only constraint used by logistic regression in maximizing entropy over the dataset, provide a natural measure of the difference in counts or rates for that feature induced by a given exposure. We report these values in the figures of the main text and in *Figure 4—figure supplements 1–67*.

The beta coefficients of the features in a logistic regression have also an intuitive interpretation, since the logarithm of the odds of being in the exposed class *C* versus the unexposed one, given the mutational data (counts or rates), is given by

$$\log \frac{p(C = exposed | X = \boldsymbol{x})}{p(C = unexposed | X = \boldsymbol{x})} = \beta^T \boldsymbol{x} \,.$$

Therefore, $e^\beta$ of a feature is the factor by which the odds of being in the exposed class increase for every extra unit increase in that feature, when all other features are kept constant. The β coefficients of the mutational signatures for each factor (aging or exposure) can be found in *Supplementary file 2* and are depicted in *Figure 4—figure supplements 1–67*.

### Prediction via logistic regression (Prediction)

We used Logistic Regression (LR) to test the predictive accuracy of each set of features representing a mutational signature as measured by AUC. We also report the performance of Linear Discriminant Analysis (LDA), and Random Forest (RF), when applied to both feature selection and prediction (*Supplementary file 1*). In both LR and LDA models, the mean vectors equal the empirical mean vector. In addition, LDA also accounts for the dependencies among the features. All methods yielded relatively comparable results in cross-validation.

### Training and testing, apparent vs cross-validated

Training was performed via five iterations of threefold cross-validation as described above. Once the predictive features had been selected, the algorithm was then trained using LR over the whole dataset for the *apparent* results, or via five iterations of fivefold cross-validation for the *cross-validated* results, where we reported the mean AUC across the five iterations.

### Speed benchmark

To assess the feasibility of implementation of the SuperSigs methodology, we provide as a performance metric the runtimes for both whole exome and whole genome sequencing datasets in *Figure 3—figure supplement 38*. Plotted runtimes start from a table of summarized mutations (each row is a patient with # of mutations per trinucleotide). As it can be observed the runtime naturally depends on the number of samples analyzed. It is, however, relatively fast at an average 27 min per dataset.

### Comparison of performance between unsupervised, SuperSigs, and randomly generated peak signatures

### Generating random signatures based on prior knowledge

When prior literature has established a strong relationship between an exposure and a particular mutational feature, that is [C>T]G for aging and C>A for smoking, we would like to know that any new candidate signatures actually improve on these central, *peak* feature. Specifically, we would like

to assess the value of the aging (Signature #1) and smoking unsupervised signatures in *Alexandrov et al., 2015*; *Alexandrov et al., 2016*; *Alexandrov et al., 2013a*; *Alexandrov et al., 2013b*, as well as of our SuperSigs, beyond the main 'peaks' already known from prior knowledge, that is [C>T]G for aging and C>A for smoking. This essentially corresponds to evaluate if the part of the distribution of an unsupervised or supervised mutational signature that is not the mutational 'peak' adds any value, according to some measure of performance (prediction or correlation).

To do this, we generate a signature for smoking, whose property is a higher proportion of C>A mutations than the other mutation types and where, beside this 'peak' at C>A, the proportion of all the other mutation types is assigned randomly. Similarly we generate a signature for aging, whose property is a higher proportion of [C>T]G mutations than the other mutation types and where, beside this 'peak' at [C>T]G, the proportion of all the other mutation types is assigned randomly. We do this by building 'randomly generated single peak signatures', or '*single peak signatures*' for brevity.

More precisely, for the smoking signature, we create this randomly generated smoking peak signature in a two-step process. In step one, we generate 30 (since in Cosmic v.2 there are about 30 signatures) probability distributions over the six main mutation types (which lack suffix and prefix base). Each distribution is created by sampling six numbers from a uniform distribution and by dividing them by their sum. The '*smoking single peak signature*' is then the distribution among them with the highest proportion of C>A substitutions. In step two, we randomly break down the obtained proportion of each of the six main mutation types into the 16 fundamental trinucleotide mutations (16 for C>A, 16 for C>T, and so on).

We apply a similar process to the derivation of the randomly generated peak age signatures. The difference is that we assume the main types of mutations are now seven: [C>T]G, [C>T]H, C>A, C>G, T>A, T>C, and T>G, due to the fact that we need to have [C>T]G as one of the features, since that is the peak obtained from prior-knowledge. Among the 30 signature candidates, the '*aging single peak signature*' is then the distribution with the maximum proportion of [C>T]G substitutions.

## Comparison of Alexandrov et al., randomly generated peak signatures, and SuperSigs

In order to compare the prediction accuracy (AUC) of all three sets of signatures (Alexandrov et al., single peak, and SuperSigs), we first apply the same prediction methodology that was previously used in Alexandrov et al. to determine the contribution of each signature in each patient: non-negative least squares (NNLS).

More specifically, to determine in a given patient the respective proportional contributions (used as a score) $X_i$ of each mutational signature $i = 1,\ldots, k$, where a total of $k$ signatures are present in that tissue, NNLS is applied to

$$Y_i = A_{i1}\ X_1 + A_{i2}\ X_2 + \ldots + A_{ik}\ X_k$$

that is $Y = AX$ in matrix form, where $Y_i$ is the total number of mutations of type $i$, and $A_{ij}$ is the relative frequency (for Alexandrov et al. and single peak signatures) or the difference in mean count (SuperSigs for age) or rate (SuperSigs for all other etiological factors) of mutation type $i$ in the mutational signature $j$, across each one of the $k$ signatures present in that tissue.

The performance of the various methodologies is presented in *Figure 3*, *Figure 3—figure supplement 1*, and *Supplementary file 1*.

For Alexandrov et al. their Signature one was used for predicting age in one comparison (*Figure 3b*), and the combination of the 'clock-wise' unsupervised Signatures 1 and 5 as determined in *Alexandrov et al., 2015* was used in the other comparison. The specific combination of signatures used for Alexandrov et al. in predicting smoking status was instead determined by the specific combinations provided for each tissue in *Alexandrov et al., 2016*. The accuracies (AUC) of all predictions can be found in *Supplementary file 1*.

## Comparison of cross-validated NMF versus SuperSigs

It was not possible to cross-validate directly the unsupervised method of *Alexandrov et al., 2013a*, due to the fact that there are some manual or semi-automated steps in their signature creation, validation and association to the risk factors. For example, in their section 'Samples and curation of

freely available cancer data' it is stated that: "*All mutational signatures were clustered using unsupervised agglomerative hierarchical clustering and a threshold was selected to identify the set of consensus mutational signatures. Mis-clustering was avoided by manual examination (and whenever necessary re-assignment) of all signatures in all clusters. 27 consensus mutational signatures were identified across the 30 cancer types.*" Thus, we chose to use the core methodology used in Alexandrov et al., which is non-negative matrix factorization (NMF) (*Lee and Seung, 1999*), and approximate their method in two alternative ways in order to perform cross-validation: (1) 'BestNMF' and (2) 'MatchedNMF', as described below.

For both approaches, we applied NMF (Lee-Sung NMF_0.21.0) to the profile of the count mutations of the training samples, that is a matrix whose 96 rows represent mutation types and columns represent training samples. We set the rank parameter, $r$, of the NMF algorithm equal to what shown in Cosmic signature v2 (https://cancer.sanger.ac.uk/cosmic/signatures_v2) for the tissue of interest. We hardwired this parameter to help the unsupervised method to limit model misspecification (see also the model misspecification section below).

After obtaining the $r$ signatures from NMF, we used two alternative methods to select among them the signature of a specific age or environmental factor: (1) for BestNMF, we chose the signature whose contributions had the highest AUC in classifying exposure to the environmental factor on the training set; (2) for MatchedNMF, we paired each of the identified signatures from the training set to exactly one of those listed in Cosmic v2 for this specific tissue. This pairing process was obtained by maximizing the sum of the cosine similarity for each pair.

Then, on the test set, we used an NNLS algorithm to estimate the contribution of each signature on the test set.

The performance of the various methodologies is presented in *Figure 3*, *Figure 3—figure supplement 1*, and *Supplementary file 1*.

We would like to note that we are being conservative against our own supervised results when using the alternative method called 'BestNMF' to cross-validate the unsupervised approach. In fact, BestNMF is more of a weakly supervised method rather than a purely unsupervised method, and therefore having an expected better performance. This can also be verified by comparing the apparent performances of Alexandrov and BestNMF in *Supplementary file 1*. Also, BestNMF provides us with signatures for cases that Alexandrov et al did not study like BMI, alcohol, etc. and it is not clear how we can generate those signatures in Alexandrov as the Cosmic signatures are pre-fixed for tissues. The closest we can get to the Alexandrov implementation is with what we termed 'MatchedNMF'. It is still in favor of the Alexandrov implementation because the process for matching the signature uses the Cosmic signatures and those signatures indirectly used the exposure labels of test labels to identify the associated risk factor as well.

## Partially supervised method extension

We explain here how the supervised learning of a mutational signature (specifically the aging signature in this example) can be used to improve the performance of an unsupervised approach by discounting the effects of that supervised signature on the test data. We refer hereafter to this methodology as 'partially supervised'.

To simplify matters, we do not engineer features; rather, we use the 96 fundamental mutations as in *Alexandrov et al., 2013b*. We only use the datasets that show a higher average rate of mutation per year in the exposed samples than in the unexposed samples. This increase in the rate is required to conform to the premise of non-negativity and linearity in the NMF model. We use one half of the unexposed samples as the training set to learn the rate of each feature of the age signature (thus a supervised signature) so that we can discount the effect of age (i.e. controlling for age) on the test set. Next, we formed the test set by bootstrapping over the left-out half of the unexposed samples and all exposed ones.

We apply NMF (Lee-Sung ref) with rank equal to three to decompose the test set, Y, thus obtaining two matrices, A and X: one containing the unsupervised signatures (A) and a second one with the corresponding contributions of each of those signatures in each patient (X). These contributions have not been discounted for age yet. This is the standard unsupervised approach. However, in order to estimate the discounted contributions of a signature in each test sample, we now discount the effect of age of a patient on each unsupervised signature, by multiplying the learned supervised age signature by the age of the patient, times the estimated mutation rate, and then projecting this

vector onto the directions identified by NMF using NNLS, and then subtracting these projected contributions of age from the contributions of the three unsupervised signatures obtained by NMF. To conform to the premises of NMF, we set the negative discounted contributions to zero.

We then choose the direction whose contribution, divided by the total number of mutations, is the most associated (in terms of the highest AUC) to the exposure status using the known ground-truth, for both the unsupervised and the partially supervised methods, by using the not discounted and discounted contributions, respectively. To obtain the 'partially supervised signatures' we now use again non-negative linear regression but this time where the contributions (X) are known and the signatures (A) are unknown. In other words, we still have the decomposition Y=AX, but now, Y and X are known and we want to estimate A.

We used the AUC to evaluate the association of the signature with the exposure status, for both the unsupervised and partially supervised approach, where the contribution of each signature has been divided by the number of total mutations. We repeated 50 times this whole process (from the random selection of half of the unexposed patients used to learn the age signature and so on) and take the average AUC over them to account for the effect of randomness. This is what is depicted in *Figure 3—figure supplement 36*, where the increase in performance of the partially supervised method with respect to the unsupervised is evident.

In this partially supervised extension, we have used NMF to easily compare with the unsupervised approach by Alexandrov et al. However, other methodologies (e.g. a classifier based on EM) may provide even better performance.

## The effect of model misspecification on the unsupervised signatures

Pretend we had no annotation for the presence of defects in the gene POL-ε among patients with endometrial cancer in the UCEC-TCGA dataset and that we did not know the POL-ε signature. The normalized results for an NMF decomposition are depicted in *Figure 3—figure supplement 37a*. This example illustrates a critical weakness of unsupervised approaches in general. The POL-ε signature in *Figure 3—figure supplement 37a* was obtained by 'telling' NMF to search for one (i.e. rank = 1) pattern. If instead we had asked for two, three, or four signatures, respectively, NMF would have returned the patterns depicted in *Figure 3—figure supplement 37b–d*. *Figure 3—figure supplement 37b–d* show that the POL-ε signature has been parsed into multiple patterns: the more patterns the more the optimum signature is spread across different claimed signatures. Therefore, the quality of the results of NMF strongly depend on the number of signatures NMF is required to extract. Unfortunately, there is no fully satisfactory rule to determine a priori how many patterns should be found by NMF. This is a problem that all unsupervised approaches have because the researcher is blind to the actual number of different exposures that are present among the patients in the dataset during the discovery phase. As Hastie et al. put it in their classic treatise: 'it is difficult to ascertain the validity of inferences drawn from the output of most unsupervised learning algorithms', and their 'effectiveness is a matter of opinion and cannot be verified directly' (p. 487 of (21)). The problem may be so severe that at times it may be best to just consider the distribution of mutation types without using NMF at all.

## Estimation of the proportion of mutations due to aging

Each predictive feature of our SuperSigs can be represented by its rate. For age, the '*rate*' of feature $i$, $r_i^a$, is defined as the mean of the ratio:

$$r_i^a = \frac{mean(count\ of\ feature\ i)}{mean(age)}$$

in unexposed patients. This rate estimates the number of mutations of that particular feature accumulating per year and attributable to age. To estimate the proportion of mutations due to aging in each specific sample, we multiplied $r_i^a$ of each feature $i$ present in our SuperSig age signature by the patient's age of that specific sample. The number obtained by summing the above counts for each feature in the age SuperSig is then divided by the total number of mutations observed in that sample. This resulting ratio, being forced to be not greater than 1, is our estimate for the proportion of somatic mutations attributable to age in that sample (see *Supplementary file 3*).

## Distances among mutational landscapes of different exposures in tissues

We define the *mutational landscape* of an exposure in a tissue as the 96-long vector (96 trinucleotide mutations) where each entry is given by the average count of that mutation type in the cohort of the samples with that exposure divided by the average age in that cohort. The mutational landscape of aging is obtained in the same way using the cohort of samples without any known exposure ('unexposed'). Then, the *distance* between any two mutational landscapes is given by the Pearson's correlation between the two mutational landscapes (see *Figure 5* in the main text and *Figure 3—figure supplements 2–18*). For the results in *Figure 3—figure supplements 19–35* the effect of age has been removed from the mutational landscape of all exposures but age, by subtracting the mutational landscape of age from the relevant exposed tissue. Replacing the distance based on correlation with one based on cosine similarity yields equivalent results.

## Robustness analysis with respect to mislabeling

To assess the robustness of the methodology with respect to the quality of the clinical annotation, we switch the labels from unexposed to exposed (or vice versa) for 5, 10, 20, and 25% of the samples in the training set. For example, non-smokers would be mislabeled as smokers and vice versa. Then we rerun our supervised method, including feature engineering and selection, on the training set to obtain new signatures. These new signatures are then used for prediction in the test set, where we used the original labels as the ground truth. The performance is reported in *Supplementary file 5*. We compare AUCs at the different mislabeling percentages and found that our method still outperforms the unsupervised method all the way up to a mislabeling proportion of 25%.

Heatmap of the distance, in terms of correlation, between any two etiological factors' mutational landscapes in the corresponding tissues. Distance not discounted for age. The *distance* between any two mutational landscapes is given by 1- the Pearson's correlation between the two mutational landscapes. See Materials and method section for details.

Heatmap of the distance, in terms of correlation, between any two etiological factors' mutational landscapes in the corresponding tissues. Distance not discounted for age. The *distance* between any two mutational landscapes is given by 1- the Pearson's correlation between the two mutational landscapes. See Materials and method section for details.

## Materials and correspondence

Correspondence and requests for materials should be addressed to C.T.: ctomasetti@jhu.edu. A repository for the R package containing the SuperSigs algorithm can be found at https://github.com/TomasettiLab/superSigs (*Kuo and Zhang, 2020*; copy archived at swh:1:rev:7d6aac85a1b3930cb93e0810039db4d65a242cca).

## Acknowledgements

We thank Dr. Henry B Laufer for critical review and useful suggestions and comments, and Mahnoosh Mehrabani for useful comments. This research was supported by The John Templeton Foundation, the Lustgarten Foundation for Pancreatic Cancer Research, The Virginia and D K Ludwig Fund for Cancer Research, and NCI grants P30CA006973.

## Additional information

### Competing interests

Ken W Kinzler: KWK is a founder of and hold equity, and serve as consultant to Thrive Earlier Detection and Personal Genome Diagnostics. KWK is on the Board of Directors of Thrive Earlier Detection. KWK is a consultant to Sysmex, Eisai, and CAGE Pharma and hold equity in CAGE Pharma. KWK is a consultant to and hold equity in NeoPhore. The companies named above, as well as other companies, have licensed previously described technologies from Johns Hopkins University. KWK is an inventor on some of these technologies. Licenses to these technologies are or will be associated

with equity or royalty payments to the inventors as well as to Johns Hopkins University. Patent applications on the work described in this paper have or may be filed by Johns Hopkins University. The terms of all these arrangements are being managed by Johns Hopkins University in accordance with its conflict of interest policies. Bert Vogelstein: BV is a founder of and hold equity, and serve as consultant to Thrive Earlier Detection and Personal Genome Diagnostics. BV is a consultant to Sysmex, Eisai, and CAGE Pharma and hold equity in CAGE Pharma. BV is also a consultant to Nexus, and is a consultant to and hold equity in NeoPhore. The companies named above, as well as other companies, have licensed previously described technologies from Johns Hopkins University. BV is an inventor on some of these technologies. Licenses to these technologies are or will be associated with equity or royalty payments to the inventors as well as to Johns Hopkins University. Patent applications on the work described in this paper have or may be filed by Johns Hopkins University. The terms of all these arrangements are being managed by Johns Hopkins University in accordance with its conflict of interest policies. Cristian Tomasetti: CT is a consultant to Bayer and Johnson & Johnson. Thrive Earlier Detection has licensed previously described technologies from Johns Hopkins University. CT is an inventor on some of these technologies. Licenses to these technologies are or will be associated with equity or royalty payments to the inventors as well as to Johns Hopkins University. Patent applications on the work described in this paper have or may be filed by Johns Hopkins University. The terms of all these arrangements are being managed by Johns Hopkins University in accordance with its conflict of interest policies. The other authors declare that no competing interests exist.

## Funding

| Funder | Grant reference number | Author |
|---|---|---|
| The John Templeton Foundation | #61471 | Bahman Afsari<br>Albert Kuo<br>YiFan Zhang<br>Lu Li<br>Kamel Lahouel<br>Ludmila Danilova<br>Cristian Tomasetti |
| Lustgarten Foundation for Pancreatic Cancer Research | 90081420 | Ken W Kinzler<br>Bert Vogelstein<br>Cristian Tomasetti |
| Virginia and D.K. Ludwig Fund for Cancer Research | | Ken W Kinzler<br>Bert Vogelstein |
| National Cancer Institute | P30CA006973 | Cristian Tomasetti |
| Russian Foundation for Basic Research | 17-00-00208 | Alexander Favorov |

The funders had no role in study design, data collection and interpretation, or the decision to submit the work for publication.

## Author contributions

Bahman Afsari, Data curation, Formal analysis, Validation, Investigation, Visualization, Methodology, Writing - original draft, Writing - review and editing; Albert Kuo, Formal analysis, Visualization, Methodology, Writing - review and editing; YiFan Zhang, Data curation, Methodology; Lu Li, Ludmila Danilova, Formal analysis; Kamel Lahouel, Alexander Favorov, Methodology; Thomas A Rosenquist, Arthur P Grollman, Ken W Kinzler, Leslie Cope, Writing - review and editing; Bert Vogelstein, Supervision, Investigation, Writing - original draft, Writing - review and editing; Cristian Tomasetti, Conceptualization, Resources, Formal analysis, Supervision, Funding acquisition, Validation, Investigation, Visualization, Methodology, Writing - original draft, Writing - review and editing

## Author ORCIDs

Albert Kuo (ID) https://orcid.org/0000-0001-5155-0748

Kamel Lahouel (ID) http://orcid.org/0000-0002-4339-5749

Ludmila Danilova ⓘ http://orcid.org/0000-0003-2813-3094
Cristian Tomasetti ⓘ https://orcid.org/0000-0003-3277-4804

**Decision letter and Author response**
Decision letter https://doi.org/10.7554/eLife.61082.sa1
Author response https://doi.org/10.7554/eLife.61082.sa2

## Additional files

**Supplementary files**
• Supplementary file 1. Comparisons of prediction accuracy (AUC) and correlation across methods. The AUCs and correlations, both apparent and cross-validated, are reported for age and all other etiological factors across all tissue types for each one of the mutational signature methodologies considered in this study: Logistic Regression (Logit), Linear Discriminant Analysis (LDA), Non-negative Least Square Logit using the Betas (NNLS_Logit_betas), Non-negative Least Square Logit using the means (NNLS_Logit_means), Random Forest (RF), Unsupervised as in *Alexandrov et al., 2013a* (Unsupervised), Best_NMF, Matched_NMF, Signature one as in *Alexandrov et al., 2013b* (Signature1), and Single Peak (SinglePeak). For their detailed description see the Materials and methods.

• Supplementary file 2. SuperSigs and their predictive features. The set of n predictive features forming the supervised signature (SuperSig) are listed for each tissue type and for each etiological exposure. Two values are associated to each one of these predictive features: (1) the difference in mean counts (age) or rates (all other exposures) between the exposed and unexposed cohorts, and (2) the beta (β) coefficient for that feature as estimated by logistic regression. See *Figure 4—figure supplements 1–67*.

• Supplementary file 3. Estimated contributions of the age signature in different tissue types. For each tissue type and for each etiological factor the estimated mean and median contribution of that factor, out of the total number of mutations present in that tissue, are reported together with the sample sizes (number of patients analyzed).

• Supplementary file 4. Comparisons of prediction accuracy (AUC) after random shuffling across methods. After random shuffling, the AUCs, both apparent and cross-validated (CV), are reported for age and all other etiological factors across all tissue types for each one of the mutational signature methodologies considered in this study: Logistic Regression (Logit), Linear Discriminant Analysis (LDA), Non-negative Least Square Logit using the Betas (NNLS_Logit_betas), Non-negative Least Square Logit using the means (NNLS_Logit_means), Random Forest (RF), Unsupervised as in Alexandrov et al. (Unsupervised), Best_NMF, Matched_NMF, Signature one as in Alexandrov et al. (Signature1), and Single Peak (SinglePeak). For their detailed description see Materials and methods.

• Supplementary file 5. Comparisons of prediction accuracy (AUC) with different mislabeled proportions (5, 10, 20, and 25% of samples mislabeled) in the training set. The AUCs, both apparent and cross-validated (CV), are reported for age and all other etiological factors across all tissue types for each one of the mutational signature methodologies considered in this study: Logistic Regression (Logit), Linear Discriminant Analysis (LDA), Non-negative Least Square Logit using the Betas (NNLS_Logit_betas), Non-negative Least Square Logit using the means (NNLS_Logit_means), Random Forest (RF), Unsupervised as in Alexandrov et al. (Unsupervised), Best_NMF, Matched_NMF, Signature one as in Alexandrov et al. (Signature1), and Single Peak (SinglePeak). For their detailed description see Materials and methods.

• Transparent reporting form

**Data availability**
All data generated during this study are included in the manuscript and supporting files. A repository for the R package containing the SuperSigs algorithm is found at https://github.com/Tomasetti-Lab/superSigs (copy archived at https://archive.softwareheritage.org/swh:1:rev:7d6aac85a1b3930cb93e0810039db4d65a242cca/).

The following datasets were generated:

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
