## [Decision Letter]

**Acceptance summary:**

The approach described in this manuscript provides a new method to analyze sequencing data, which will be of general use to the scientific community. In addition, this work identifies hypotheses concerning the etiology of certain cancers.

**Decision letter after peer review:**

Thank you for submitting your article "Supervised mutational signatures for obesity and other tissue-specific etiological factors in cancer" for consideration by *eLife*. Your article has been reviewed by four peer reviewers, one of whom is a member of our Board of Reviewing Editors, and the evaluation has been overseen by a Senior Editor. The following individuals involved in review of your submission have agreed to reveal their identity: Peter J Campbell (Reviewer #2); Elaine Mardis (Reviewer #4).

The reviewers have discussed the reviews with one another and the Reviewing Editor has drafted this decision to help you prepare a revised submission.

Summary:

This manuscript by Tomasetti and colleagues describes a conceptually novel approach to derive somatic mutational signatures (SuperSigs) from whole exome tumor sequencing data using supervised machine-learning techniques. A large and growing number of mutational signatures have been described using non-negative matrix factorization (NMF), an unsupervised method. These signatures are widely used to describe the mutational processes operant in individual tumors and have spawned many studies examining environmental exposures and intrinsic genetic mechanisms that underlie these patterns. However, when accompanying clinical data are available, supervised learning may produce more informative signatures that correlate more strongly with environmental exposures, as well as enable the discovery of signatures associated with previously unappreciated factors. This work describes one such effort and using these methods, identify differences in penetrance of such signatures in a tissue restricted manner.

Essential revisions:

1) One weakness is the reliance on one dataset, and a highly curated, highly polished one at that. The authors used the TCGA data with random splits to training and test sets via a cross-validation design. The vast majority of the data studied was exome data, with only AML and ovarian cancers using whole genome data. A specific challenge of supervised machine learning techniques is that any hidden confounding factor in the dataset (say, hospital of origin, sequencing centre, variant calling) can be extracted as apparent signal for the variable of interest (say age, obesity) – a random test / training split on one dataset will still be subject to this confounding, since it operates at the level of the combined patients. The best way to avoid this (although it is not failsafe either) is to validate signal on external datasets.

We therefore recommend testing predictors trained on TCGA data but tested on external data. There is plenty of non-TCGA exome data available and accessible, and even reasonable amounts of whole genome data, with basic clinical variables (smoking, age, sex for sure; obesity more patchy) – this would give much greater confidence in the signals extracted.

2) The authors make reference to "partially-supervised signatures" in Figure 2A and in the text, claiming that they are "superior to" unsupervised signatures when clinically annotations are not available. This is confusing because SuperSigs is motivated by the advantage of supervised methods specifically for when clinical data are available. Moreover, I don't see any data presented in the Results to support the claim that partially-supervised signatures are better. While there is a detailed description in Materials and methods, more description and evidence should be included in the text, or else the paper should focus only on the scenario when clinical data are used.

3) The figures are generally unclear and need to be improved. For instance, Figure 2B attempts to show examples of randomly generated single peak signatures, but I am confused why there are multiple colored dots for each trinucleotide if there is only one signature per color. I believe the trinucleotide sequences are also incorrect, as the right half of the plots in 2B should have a T at the center of each trinucleotide, not C. Additionally, it is unclear in Figure 3 (panel B) which tumor types were chosen for the characterization of smoking signatures, and why. I also can't understand why smoking is shown again in panel c or why age is shown in panel d when the legend describes it as "all etiologic factors other than age".

4) The Introduction mentions a critique of unsupervised NMF signatures that they don't incorporate knowledge of exposures or their intensity (e.g., cigarettes packs/day). But it is not clear that SuperSigs explicitly models the intensity either. Samples are divided into two categories: "unexposed" and "exposed". Is the intensity of smoking exposure reflected in the smoking signatures from different tissues?

5) How many tumor types have a detectable smoking signature? How many tumor types were examined for a smoking signature? Figure 4 displays smoking signatures for pancreatic adenocarcinoma and bladder urothelial carcinoma, which are surprising given that these tissues do not have nearly as much direct exposure to the carcinogens in cigarette smoke as lung cancer, head and neck cancer, and esophageal cancer.

6) The authors fail to address the potential for overfitting. A strong test against overfitting would be to shuffle the labels and then re-run their entire framework from feature selection to test prediction. A clear sign the model is overfitting would be obtaining similar AUCs on the shuffled dataset to that of their unshuffled dataset.

7) AUCs as a metric for model performance can be deceiving; an H-measure would be more informative here see David J. Hands paper 2009 Machine Learning for more details.

8) The addition of performance metrics such as speed benchmarks for exome and WGS data sets is needed to assess the feasibility of implementation for end users. This is especially true considering the large number of bootstraps needs for feature engineering.

9) How does the novel feature engineering technique compare to state-of-the-art methods like sigLASSO? Does it take into account the transcript strand bias of genes?

10) Supervised learning depends on the quality of the data annotations, which as noted above are quite variable for different TCGA projects. In most instances, there are not quantitative information available to establish what age ranges were considered “young” vs. “old”, nor were there specifics around environmental exposures such as pack-years for smoking, ranges of BMI for obesity, etc. Could these please be specified and also some reference as to how these were selected, and were they selected prior to or after cross-validation was performed for the different categories?

11) The conclusions about SuperSigs algorithm performance compared to NMF seem to be based on core NMF rather than the improved versions from Alexandrov. The authors claim that it is not possible to use Alexandrov's implementation, however, offer no reason as to why. Please specify the NMF approach used in the Materials and methods so it is clear to readers.

12) The authors provide a strong rationale for the greater flexibility for mutational signatures beyond a three nucleotide fixed size improves their models' performance, but don't provide statistical analysis to support this.

13) Context of this publication with previous work. The aging signature is based on mutation counts, rather than rates. Does this not vary according to the tissue, based on the prior work published by this group? Is this another basic principle that could be addressed by comparing to SuperSigs analysis of pediatric cancer data?

14) The conclusions around obesity and tissue variance are overstated with respect to causality. Since these studies are observational, the authors cannot conclude evidence of causality.

[Editors' note: further revisions were suggested prior to acceptance, as described below.]

Thank you for resubmitting your revised article "Supervised mutational signatures for obesity and other tissue-specific etiological factors in cancer" for consideration by *eLife*. Your revised article has been re-reviewed by two peer reviewers, and the evaluation has been overseen by a Reviewing Editor and a Senior Editor. The following individual involved in review of your revised submission has agreed to reveal their identity: Elaine Mardis (Reviewer #4).

As for the original submission, the reviewers have discussed their reviews with one another, and the Reviewing Editor has drafted this to help you prepare a revised submission.

Summary:

In this manuscript, the authors describe a new approach to identify patterns of genetic alterations in tumors that reflect biological processes that affect DNA. This approach does not use prior information and uses more sequence information to look for evidence of mutational processes. Among many interesting observations is the finding that certain mutational processes occasionally have different signatures across tissue types.

Essential Revisions:

1) The authors should revise the description of the methods to provide greater detail. One example of this is the last paragraph in subsection “Do mutational signatures add to prior knowledge about etiologic factors?”, where "information" is used extensively throughout, without ever really defining what type of information is being provided by SuperSigs.

2) Please provide a weblink for the TCGA Genomics Data Commons.

3) In subsection “Do mutational signatures add to prior knowledge about etiologic factors?” (and throughout), resist using "significantly" unless you have a p value that substantiates the term, and the test used to derive the p value.

4) In the Discussion, it is mentioned that SuperSigs associated with BRCA2 vary with tissue type (ostensibly meaning breast and ovarian) but only SuperSigs for breast cancer associated with BRCA2 is shown in Figure 4.

5) Discussion paragraph two is the first mention of inflammation, which is known to be linked to obesity yet is not specifically called out here.

6) In the same paragraph, both signature 4 and signature 6 are mentioned in terms of liver cancer but only signature 6 is explained as being “rare in liver cancers”. is it also the case that smoking is not associated with liver cancer, or…? Please edit this sentence so the reader understands why you are also mentioning the smoking signature result.

Reviewer #3:

This manuscript describes a conceptually novel approach to derive somatic mutational signatures (SuperSigs) from whole exome tumor sequencing data using supervised machine-learning techniques. A large and growing number of mutational signatures have been described (initially by Alexandrov et al) using non-negative matrix factorization (NMF), an unsupervised method. These signatures are widely used to describe the mutational processes operant in individual tumors and have spawned many studies examining environmental exposures and intrinsic genetic mechanisms that underlie these patterns. However, when accompanying clinical data are available, supervised learning may produce more informative signatures that correlate more strongly with environmental exposures, as well as enable the discovery of signatures associated with previously unappreciated factors.

This study is innovative, clearly written, and logically presents the benefits of supervised learning methods to identify signatures more strongly associated with meaningful clinical factors. The conceptual advantages of SuperSigs as an alternative to NMF-based signatures are compelling. Among many interesting observations is the finding that certain mutational processes occasionally have different signatures across tissue types. Given the ubiquity of mutational signatures in cancer genomics research and the challenge in connecting many signatures to their underlying cause, I think this study and the associated algorithm will be of broad interest.

Recommendations for the authors:

I am satisfied that the authors have sufficiently addressed the concerns raised in my original review.

Reviewer #4:

This manuscript describes an approach to uncovering mutational signatures and their associated causative factors, using a new approach that does not rely on NMF/unsupervised methods, which are in widespread use yet often yield complex or indeterminate underlying causes. The work takes advantage of publicly available TCGA genomic data and associated clinical data for 30 different cancer types, and utilizes the ICGC data to test their supervised approach to mutational signatures, or SuperSigs. Important results obtained by this approach include the observation that signatures of aging differ between different tissue types, perhaps linked to the underlying cell division rate differences that are well known to exist. Another important result is the derivation of two signatures related to obesity, which is emerging as the key contributor to cancer susceptibility in first world economies. Another major strength of this approach is that by first removing known etiological signatures such as smoking with SuperSigs, more refined signatures of underlying or associated etiologic factors may be revealed by a secondary NMF-based approach. This partially supervised method achieves higher concordance than by NMF only for all known etiologies (genetic and environmental) recorded in TCGA clinical data. Of necessity, this approach was tested with a large data set that left out the nuances of different tumor types. For example, breast cancers are well characterized into different subtypes, especially with respect to underlying factors such as BRCA1/2 mutation status (germline) that associate with the different subtypes (e.g. triple negative disease). As such, it remains to be seen whether this method may be more specifically applied to large genomic data sets across different subtypes and in the context of underlying genetic predispositions to further understand the susceptibility-based etiology of a specific tissue type. However, as a general method, SuperSigs provides a new and well-tested method for deriving mutational signatures from cancer genomic data that likely will have a significant impact on the field of cancer research and, ultimately in areas of cancer prevention as well as revealing new cancer etiologies.

Recommendations for the authors:

In general, the authors responded well to the critiques raised, including the ones I contributed. I was pleased they were able to utilize the ICGC data and to show the method transferred well to the analysis of these data. It would be a good idea for the figure they produced in the response to reviewers to be included as a supplementary figure in the revised manuscript, if possible.

The figures are all quite improved, although the use of colors to help differentiate different tumor types and different etiologies sometimes is challenging to decode. This will likely be helped by preparation for publication. My only remaining critique is the language used to quantify the SuperSigs improvements over NMF are sometimes poorly quantitative. One great example of this is the last paragraph in subsection “Do mutational signatures add to prior knowledge about etiologic factors?”, where "information" is used extensively throughout, without ever really defining what type of information is being provided by SuperSigs. This needs to be edited accordingly. Please provide a weblink for the TCGA Genomics Data Commons. In subsection “Do mutational signatures add to prior knowledge about etiologic factors?” (and throughout), resist using "significantly" unless you have a p value that substantiates the term, and the test used to derive the p value. This is standard. In the Discussion section, there are two areas of confusion for me. In the Discussion, it is mentioned that SuperSigs associated with BRCA2 vary with tissue type (ostensibly meaning breast and ovarian) but I only found a SuperSigs for breast cancer associated with BRCA2 in Figure 4. This is a very important observation, so please be certain it is presented for both tissues in an accessible way in the main text. Discussion paragraph two is the first mention of inflammation, which is known to be linked to obesity yet is not specifically called out here. If this is indeed the inference being made, please be more explicit so the reader understands the importance of this conclusion! In the same paragraph, both signature 4 and signature 6 are mentioned in terms of liver cancer but only signature 6 is explained as being “rare in liver cancers”. is it also the case that smoking is not associated with liver cancer, or…? Please edit this sentence so the reader understands why you are also mentioning the smoking signature result.

---

## [Author Response]

Essential revisions:1) One weakness is the reliance on one dataset, and a highly curated, highly polished one at that. The authors used the TCGA data with random splits to training and test sets via a cross-validation design. The vast majority of the data studied was exome data, with only AML and ovarian cancers using whole genome data. A specific challenge of supervised machine learning techniques is that any hidden confounding factor in the dataset (say, hospital of origin, sequencing centre, variant calling) can be extracted as apparent signal for the variable of interest (say age, obesity) – a random test / training split on one dataset will still be subject to this confounding, since it operates at the level of the combined patients. The best way to avoid this (although it is not failsafe either) is to validate signal on external datasets.We therefore recommend testing predictors trained on TCGA data but tested on external data. There is plenty of non-TCGA exome data available and accessible, and even reasonable amounts of whole genome data, with basic clinical variables (smoking, age, sex for sure; obesity more patchy) – this would give much greater confidence in the signals extracted.

We are grateful for this critical recommendation by the reviewers. We have added a whole new section in the main text entirely dedicated to validation, describing the results of four different validations: cross-validation, random shuffling, partial random shuffling, and testing on external data. They all provide positive support for our supervised signatures.

With respect to testing on external data, and as requested by the reviewers, we externally validated our results on an independent WGS dataset from the ICGC database (downloaded from https://dcc.icgc.org/releases/PCAWG/). As some TCGA datasets are also present in ICGC, only ICGC datasets that were not present in TCGA were used for testing. All signatures that had achieved a cross-validated AUC of at least 0.6 on TCGA data were tested, as those not satisfying this minimum requirement should not be considered reliable. The way we tested on ICGC data our predictors was as follows: using the logistic regression models trained exclusively on TCGA data, with their associated signatures, we predicted the unexposed versus exposed status for all available factors and tissues in the ICGC database. The AUC for every of those signatures is reported in Table 1 of the main text. We found that the predictive power of the signatures that was observed in the TCGA data was retained in the ICGC data. In fact, several of them achieved AUCs that were much higher than those obtained in cross-validation using TCGA data. The only exception was the age signature in melanoma (SKCM). This signature, however, had a borderline performance already in the original analysis (AUC=0.61), probably due to the inability of the methodology to properly remove sun exposure as a confounding.

2) The authors make reference to "partially-supervised signatures" in Figure 2A and in the text, claiming that they are "superior to" unsupervised signatures when clinically annotations are not available. This is confusing because SuperSigs is motivated by the advantage of supervised methods specifically for when clinical data are available. Moreover, I don't see any data presented in the Results to support the claim that partially-supervised signatures are better. While there is a detailed description in Materials and methods, more description and evidence should be included in the text, or else the paper should focus only on the scenario when clinical data are used.

It is true that our supervised methodology requires clinical annotation. At the same time there are many cases where we may have annotation available for some factors – e.g. age is typically available – but not for others. And it may be desirable to attempt to identify and detect those other non-annotated factors. Are we then forced to exclusively use an unsupervised methodology? The answer is no. The rationale for extending the method to a partially-supervised approach is that any time some annotated factors are present it is better to take care of them first, by identifying them using a supervised approach and removing their effects, and then applying an unsupervised methodology on the mutational “leftover”, rather than using the unsupervised methodology on the whole. We show this in our analysis, which we have now brought to the main text with a figure (Figure 3C) and a more detailed description. We consider it an important finding as it indicates that even when using an unsupervised approach with the intent to discover new, not annotated signatures, it is still best to first take care of the annotated component present in the mutational load of a sample.

3) The figures are generally unclear and need to be improved. For instance, Figure 2B attempts to show examples of randomly generated single peak signatures, but I am confused why there are multiple colored dots for each trinucleotide if there is only one signature per color. I believe the trinucleotide sequences are also incorrect, as the right half of the plots in 2b should have a T at the center of each trinucleotide, not C. Additionally, it is unclear in Figure 3 (panel b) which tumor types were chosen for the characterization of smoking signatures, and why. I also can't understand why smoking is shown again in panel c or why age is shown in panel d when the legend describes it as "all etiologic factors other than age".

We thank the reviewers for pointing out that some figures were unclear and needed improvements. We have redone almost all figures. Part of the confusion in Figure 2 was generated by the fact that 3 distinct random single peak signatures were represented in the lower part of Figure 2B. We have now reduced that number to only one, to make the interpretation of the figure clearer. We also changed the colors accordingly (color now corresponds to mutation type), and fix the typo in the trinucleotide sequence. In Figure 3 we have eliminated some confusion created by the fact that some factors were reported in multiple plots. In particular smoking was reported on its own as well as in other plots including all factors. We have reduced the figure to two plots, one comparing the performance of the supervised and unsupervised methods over age and one for all other factors. As indicated in the text, all TCGA datasets (tumor types) containing smoking status information (8 total) had been analyzed, and their performance reported in Supplementary file 1. We have now added the names of those 8 types in the main text for added clarity. However, only the 5 tumor types for which Alexandrov et al. found the presence of a smoking signature are reported in Figure 3B, as only for them we can present a comparison. We have clarified this further in the caption of Figure 3.

4) The Introduction mentions a critique of unsupervised NMF signatures that they don't incorporate knowledge of exposures or their intensity (e.g., cigarettes packs/day). But it is not clear that SuperSigs explicitly models the intensity either. Samples are divided into two categories: "unexposed" and "exposed". Is the intensity of smoking exposure reflected in the smoking signatures from different tissues?

Our reference to NMF not requiring knowledge of the intensity of an exposure was meant as a potential advantage of NMF and not as a criticism. We have removed the indicated sentence on intensity to avoid any confusion. In principle SuperSigs can be used to model different intensities of exposure by creating multiple groups of exposed patients clustered by intensity. We have not done that in this manuscript. However, we are reporting in the main text correlations between exposure intensities in smoking and the score obtained using both SuperSigs and NMF signatures.

5) How many tumor types have a detectable smoking signature? How many tumor types were examined for a smoking signature? Figure 4 displays smoking signatures for pancreatic adenocarcinoma and bladder urothelial carcinoma, which are surprising given that these tissues do not have nearly as much direct exposure to the carcinogens in cigarette smoke as lung cancer, head and neck cancer, and esophageal cancer.

We thank the reviewers for these questions and comment. We analyzed all tumor types for which smoking status information was available in the TCGA database, for a total of 8 tumor types, and their performance reported in Supplementary file 1. Specifically, bladder (BLCA), cervical (CESC), two esophageal (ESCAD and ESCSQ), head and neck (HNSCC), kidney (KIRP), lung (LUAD), and pancreatic (PAAD) cancers. While all these tumor types were examined for a smoking signature, only for three of them (BLCA, HSCN, LUAD) the signature satisfied a minimum requirement in cross-validation (CV AUC>0.60). For added clarity we have now made this explicit in the main text, indicating that in general any signature with a CV AUC<0.60 should not be considered reliable, and only reported those 3 smoking signatures in Figure 4C, moving the full list of smoking signatures to the supplementary materials, as they may still carry valuable information.

With respect to the two specific tissues mentioned by the reviewer, the smoking signature in pancreatic adenocarcinoma does not pass this minimum requirement, but smoking in bladder cancer does, with a CV AUC=0.68, suggesting a smaller direct exposure than in head and neck (CV AUC=0.81) or lung (CV AUC=0.90), yet still a relevant one.

6) The authors fail to address the potential for overfitting. A strong test against overfitting would be to shuffle the labels and then re-run their entire framework from feature selection to test prediction. A clear sign the model is overfitting would be obtaining similar AUCs on the shuffled dataset to that of their unshuffled dataset.

We thank the reviewers for this recommendation. As mentioned above, we have added a whole new section in the main text entirely dedicated to validation, describing the results of four different validations: cross-validation, random shuffling (i.e. the one asked here by the reviewer), partial random shuffling, and testing on external data. They all provide positive support for the robustness of our supervised signatures. Also, to minimize further the risk of overfitting, we have now slightly simplified the methodology requiring that each individual feature selected satisfies an AUC>0.60 (see Materials and methods). This has the desirable effect to increase the robustness of the signatures, reducing the drop in performance when going from apparent to cross-validated AUC, and resulting in SuperSigs with a smaller and more robust number of predictive features.

7) AUCs as a metric for model performance can be deceiving; an H-measure would be more informative here see David J. Hands paper 2009 Machine Learning for more details.

We thank the reviewer for this interesting recommendation, as we agree that AUC as a measure of performance has important limitations. At the same time, we would like to leave the reviewer suggestion to future work as the AUC is a very familiar and typical measure used in the literature and switching to an H-measure may obscure a bit the main points of this manuscripts. Also, using an H-measure would require us to make some arbitrary choices (β distribution parameters) and force us to completely modify our approach in several parts (e.g. we use AUC in feature engineering/selection) and revisit all of our results. Of course, we will be happy to implement it if required by the reviewer.

8) The addition of performance metrics such as speed benchmarks for exome and WGS data sets is needed to assess the feasibility of implementation for end users. This is especially true considering the large number of bootstraps needs for feature engineering.

We now include this information in a section of the Materials and methods with a Figure 3—figure supplement 38 depicting the runtimes for both whole exome and whole genome datasets The runtime naturally depends on the number of samples analyzed but it is relatively fast at an average of 27 minutes per dataset.

9) How does the novel feature engineering technique compare to state-of-the-art methods like sigLASSO? Does it take into account the transcript strand bias of genes?

The main message of this paper is that using clinical information – i.e. with supervision – can help us improve mutational signatures. While the sigLasso contribution is certainly an improvement over the standard methodology, it still has the flavor of an unsupervised feature engineering. It will be interesting to work on combining ideas from the sigLasso methodology with a supervised method like superSig, but we feel it is beyond the scope of this first paper. We cited this as a future direction in our Discussion section of the main text: “And more sophisticated extensions to SuperSigs obtained by borrowing advanced sparse techniques used in, e.g., the sigLassso method [1], may provide further significant improvements.” In terms of strand bias of genes, we did not use it but it is easy to see how to control for it in the future extensions (e.g. counting mutations separately by strand).

10) Supervised learning depends on the quality of the data annotations, which as noted above are quite variable for different TCGA projects. In most instances, there are not quantitative information available to establish what age ranges were considered “young” vs. “old”, nor were there specifics around environmental exposures such as pack-years for smoking, ranges of BMI for obesity, etc. Could these please be specified and also some reference as to how these were selected, and were they selected prior to or after cross-validation was performed for the different categories?

We thank the reviewer for the request. All criteria for young/old, smoker vs never-smoker, etc., were selected prior to cross-validation. To clarify, we have now included the following paragraph in “The mutational family tree” section in the Materials and methods:

“All criteria for splitting samples in young vs old, smoker vs never-smoker, etc., were selected prior to cross-validation or any application of our algorithm. For the age analysis, we divided the unexposed samples into three groups (younger, middle-aged, older) based on the tertiles of the age distribution and discarded the middle group before training the algorithm. For all other exposures, unexposed and exposed – as indicated by the available annotation – formed the two groups except for: (1) obesity, where a BMI = 30 was used as the splitting value between obese and non-obese, and (2) ultraviolet light (UV) and asbestos, for which samples with respectively the lowest 10% and 33% of the Total Mutations count were used as a proxy for the unexposed group, and all the other samples for the exposed one.”

We are also attaching in Author response image 1 a plot of the age distribution with the two tertiles considered for young and old across tumor types. We will be happy to include it in the manuscript if requested.

11) The conclusions about SuperSigs algorithm performance compared to NMF seem to be based on core NMF rather than the improved versions from Alexandrov. The authors claim that it is not possible to use Alexandrov's implementation, however, offer no reason as to why. Please specify the NMF approach used in the Materials and methods so it is clear to readers.

We appreciate the request. We have now clarified it in both the main text and in the Materials and methods section. The cross-validation of Alexandrov’s implementation was not possible mainly due to the multiple manual or only semi-automated steps in their signature creation, validation and association to the risk factors. For example, in their 2013 paper, in the section “Samples and curation of freely available cancer data” it is stated that: “All mutational signatures were clustered using unsupervised agglomerative hierarchical clustering and a threshold was selected to identify the set of consensus mutational signatures. Mis-clustering was avoided by manual examination (and whenever necessary re-assignment) of all signatures in all clusters. 27 consensus mutational signatures were identified across the 30 cancer types.” Thus, we chose to use the core methodology used in Alexandrov et al., which is non-negative matrix factorization (NMF) (Lee and Seung, 1999), and approximate their method in two alternative ways in order to perform cross-validation: (1) “BestNMF” and (2) “MatchedNMF”.

We would also like to note that we are being conservative against our own supervised results when using the alternative method called “BestNMF” to cross-validate the unsupervised approach. In fact, BestNMF is more of a weakly-supervised method rather than a purely unsupervised method. This can also be verified by comparing the apparent performances of Alexandrov and BestNMF in Supplementary file 3. Also, BestNMF provides us with signatures for cases that Alexandrov et al. did not study like BMI, alcohol, etc. and it is not clear how we can generate those signatures in Alexandrov as the Cosmic signatures are pre-fixed for tissues.

The closest we can get to the Alexandrov implementation is with what we termed “MatchedNMF”. It is still in favor of the Alexandrov implementation because the process for matching the signature uses the Cosmic signatures and those signatures indirectly used the exposure labels of test labels to identify the associated risk factor as well.

12) The authors provide a strong rationale for the greater flexibility for mutational signatures beyond a three nucleotide fixed size improves their models' performance, but don't provide statistical analysis to support this.

We apologize for any possible confusion. We could not find claims of improved performance, rather only of greater flexibility. Our current approach is indeed offering higher flexibility as the fundamental unit of analysis is not forced to be the trinucleotide, but it is found to be of length 1, 2 or 3 nucleotides depending on our statistically guided approach of feature engineering/selection (as depicted for example in Figure 4 of the main text, with signatures containing nucleotides, binucleotides and trinucleotides as features, which is different from Alexandrov’s). Thus, the SuperSigs already contain features of variable length in the current manuscript. The only thing we have not explored in the current work is lengths > 3. While one of the potential strengths of our methodology is that it naturally allows for its extension to greater lengths, we feel that the most important aspects of our approach is the flexibility obtained in the choice of a given feature length within a given maximum length. We have now clarified further in the main text that “We will restrict our present analysis to only 1, 2, and 3 base pairs lengths to simplify the presentation, as this is already sufficiently different from current methods using only trinucleotides, and leave the further extension to future work.”

13) Context of this publication with previous work. The aging signature is based on mutation counts, rather than rates. Does this not vary according to the tissue, based on the prior work published by this group? Is this another basic principle that could be addressed by comparing to SuperSigs analysis of pediatric cancer data?

The reviewer makes an important point: counts vary with tissue. We plan to address this in a follow-up manuscript, as in this one we have focused and addressed a somewhat related point: the relative contribution of the normal aging process to the overall mutational load.

14) The conclusions around obesity and tissue variance are overstated with respect to causality. Since these studies are observational, the authors cannot conclude evidence of causality.

We have toned down our description of the findings. For example, we have now eliminated two of the four obesity signatures on the base of a poor performance in terms of cross-validated AUC. We also have now used the word “suggest” rather than “provide” in several places to lessen the strength of the statements. In some other cases, the statements are referring specifically to the SuperSigs (e.g. “the SuperSigs associated with the same mutational factors varied across tissues,”).

[Editors' note: further revisions were suggested prior to acceptance, as described below.]

Essential Revisions:1) The authors should revise the description of the methods to provide greater detail. One example of this is the last paragraph in subsection “Do mutational signatures add to prior knowledge about etiologic factors?”, where "information" is used extensively throughout, without ever really defining what type of information is being provided by SuperSigs.

We have now provided greater detail on the method, specifically on the way we defined “information”. We have done so by including the following new paragraph defining “information” the first time we use it in the main text:

“In addition to simple performance, it is also important to evaluate the degree to which a given mutational signature improves upon prior knowledge about the mutational effects of an exposure to an etiological factor (Figure 2A). For example, consider the case when clinical annotation is available and the main “peak” of a mutational signature, i.e. its most common mutation, is already known before the mutational signature is obtained. The peak may be a nucleotide, a dinucleotide, or a trinucleotide, depending on the specific mutational process. For example, prior validated knowledge indicated that aging induces [C>T]G mutations, and smoking induces C>A mutations. The added value of a mutational signature then depends on the additional “information” that signature provides beyond this already-known peak. If a mutational signature yields additional mutations that, under the effects of a given exposure, the genome is enriched for –– but was previously unknown to be –– then that signature adds valuable information to prior knowledge. Mathematically, a mutational signature is represented by the set of “weights” that signature attributes to all mutations included in the analysis, with the larger weights associated to mutations the signature is more enriched for. If these weights enable a mutational signature to have a higher prediction accuracy, or correlation, than random weights do, then we say that mutational signature provides “information”.”

2) Please provide a weblink for the TCGA Genomics Data Commons.

Thank you, we have now added the link (it was cited in the Materials and methods).

3) In subsection “Do mutational signatures add to prior knowledge about etiologic factors?” (and throughout), resist using "significantly" unless you have a p value that substantiates the term, and the test used to derive the p value.

We thank the reviewer for spotting that mistake. We have also specified the test and its significance level. All other times a “significant” appears is either when citing results claimed to be significant by other papers (Alexandrov et al., or epidemiological studies on obesity, …), or just descriptions of a statistical test – and associated significance thresholds required – used when describing our methodology.

4) In the Discussion, it is mentioned that SuperSigs associated with BRCA2 vary with tissue type (ostensibly meaning breast and ovarian) but only SuperSigs for breast cancer associated with BRCA2 is shown in Figure 4.

We fully agree with the reviewer that this is an important observation. We have updated Figure 4, which now includes the BRCA2 signatures for both breast and ovarian cancers, to make this observation more easily accessible in the main text.

5) Discussion paragraph two is the first mention of inflammation, which is known to be linked to obesity yet is not specifically called out here.

We thank the reviewer for this important comment. We have now specifically called out obesity, in that sentence. We have also explained further the connection between our findings and the literature on inflammation (Cheah et al., 2015; Walser et al., 2008) which was already cited. The text now reads:

“These analyses then suggest that a major effect of some environmental factors may simply be to increase the rate of cell division. This increase would induce a linearly proportional increase in mutation rate, but with a mutation pattern that remains similar to the one caused by normal aging, i.e. it would not be associated with new signatures such as those caused by direct interaction of carcinogens with DNA. Increases in the rate of cell division are known to occur when tissues are damaged or inflamed (Cheah et al., 2015; Walser et al., 2008). These observed similarities between the environmental and aging signatures then support the idea that, in certain tissues, the environmental factor’s main effect is to induce inflammation in that tissue, thus increasing its cells’ division rate.”

6) In the same paragraph, both signature 4 and signature 6 are mentioned in terms of liver cancer but only signature 6 is explained as being “rare in liver cancers”. is it also the case that smoking is not associated with liver cancer, or…? Please edit this sentence so the reader understands why you are also mentioning the smoking signature result.

Thank you for the comment. We have now edited the sentence to include also smoking, as follows:

“Similarly, Signature 4, a signature for tobacco smoking, and Signature 6, a signature associated with defective mismatch repair mechanisms (MMR), was found in virtually every liver cancer patient (see Figure S43 in (Alexandrov, Nik-Zainal, Wedge, Aparicio, et al., 2013)), though it is unlikely that all kidney cancer patients included in the TCGA database were smokers and MMR-deficiency is rare in liver cancers.”

References

1) Li, Shantao, Forrest W. Crawford, and Mark B. Gerstein. "Using sigLASSO to optimize cancer mutation signatures jointly with sampling likelihood." Nature communications 11.1 (2020): 1-12.